# Energetic shifts reflect survival likelihood in *Anopheles gambiae*
Tiago G. Zeferino ©[1,3], Luís M. Silva ©[1,2,3] ✉ & Jacob C. Koella[1]

Life history theory predicts that resource allocation adapts to ecological and evolutionary pressures. We investigated resource and energy in the malaria vector *Anopheles gambiae* following exposure to two stressors: blood meals and infection by the microsporidian *Vavraia culicis*. Our findings reveal the costs of blood-feeding and parasitism on longevity, highlighting trade-offs in lifetime protein, carbohydrate, and lipid reserves. Notably, shifts in carbohydrate-to-lipid ratios were associated with survival likelihood, with survivors exhibiting higher resource reserves and uniquely transitioning from carbohydrate to lipid utilisation, a pattern absent in non-survivors. This study emphasises the coevolutionary dynamics between hosts and parasites, highlighting how intrinsic and extrinsic factors shape host physiology. More broadly, our results underscore the importance of integrating host metabolic responses into ecological and epidemiological frameworks to enhance understanding of parasite transmission and survival strategies.

According to life history theory[1,2], organisms allocate resources to traits throughout their lives to maximise their reproductive success, which leads to trade-offs among the involved traits. Traditionally, the traits considered were primarily age-specific growth rates, fecundity, and mortality rates. More recently, the approach has been expanded to study, for example, decisions about resource allocation to growth, reproduction, and immune responses theoretically[3] and experimentally[4], and empirical work has confirmed the predicted trade-offs between fecundity and immune efficacy[5,6].

While such work has made it clear that life history theory can help understand host-parasite interactions, it also shows that the approach requires more detailed data on how different resources are allocated to functions and how these allocation patterns affect traits and underlie the trade-offs among them. One limitation in linking resource allocation and parasite-host interactions is that studies of resource sequestration by parasites often focus on specific time points rather than encompassing the host's entire adulthood, as illustrated by a review of malaria parasites in *Anopheles* mosquitoes[7].

Mosquitoes are well-suited for such studies, as they obtain their primary resources from two food sources: protein and lipids from blood[8,9] and carbohydrates from the nectar of plants[10]. Proteins, carbohydrates, and lipids play diverse roles in organisms, serving as structural components, enzymatic cofactors, and energy sources. In this study, we focus on the energetic roles of carbohydrates and lipids, as well as the reproductive roles of proteins. Carbohydrates serve as an immediate energy source (e.g., glycogen), while lipids (e.g., triglycerides) provide stored energy crucial for reproduction. Proteins are essential for oogenesis, particularly in the

production of vitellogenin, a yolk precursor protein that supports embryonic development[11,12]. Blood meals, however, expose mosquitoes to several risks, including parasitism and unwanted blood components[13,14], and they trigger oxidative stress responses[15,16], damage tissues because of the abdomen extension[9,17], and activate immune responses to fend off parasitic infections[18,19].

The long-term implications of blood meals on resource dynamics and their interaction with parasitism remain poorly understood[20]. To move towards a description of the role of resource allocation on the interaction between a host's life history and parasitic infection, we investigated the impact of a blood meal and infection by the microsporidian *Vavraia culicis*[21] on the longevity and resource dynamics of the mosquito *An. gambiae*. Although *V. culicis* depletes larval resources in *Ae. aegypti*[22], it has few longevity and fecundity costs[23–25], enabling us to track long-term changes in resource allocation. In our experiment (Fig. 1), we exposed some larvae to *V. culicis* and offered adults a blood meal or not, and then: (a) monitored the dynamics of the total protein, total carbohydrate, and total lipid content; (b) measured parasite load at several time points throughout the mosquitoes' lives; and (c) evaluated a possible protective role of the concentrations of resource in mosquito survival. Through this comprehensive approach, we aimed to shed light on the influence of mosquito ecology and physiology on infection progression and outcome. Our findings showed that although resource levels throughout the adult mosquito's life were not influenced by blood meal and infection, resources were tightly linked to adult survivorship, particularly carbohydrates and lipids, suggesting a potential correlation with their mortality likelihood. This insight is crucial for advancing our

[1]Institute of Biology, University of Neuchâtel, Neuchâtel, Switzerland. [2]Department of Zoology, University of British Columbia, Vancouver, Canada. [3]These authors contributed equally: Tiago G. Zeferino, Luís M. Silva. ✉e-mail: luis.silva@ubc.ca

**Fig. 1 | Experimental design.** Illustration of the experiment in this study, including treatments (i.e., blood meal or not, and exposed or not to *V. culicis*), and alive and at death sample collection throughout *An. gambiae* lifetime. Individuals collected at death were allocated into subgroups (e.g., individuals that died between days 4.5 and 9.5 were assigned to age class 7, while those that died between days 9.5 and 14.5 were assigned to age class 12, and so on) so that they could be compared to the individuals collected alive. Individuals in blood-meal treatments had their meal on day 7 of adulthood.

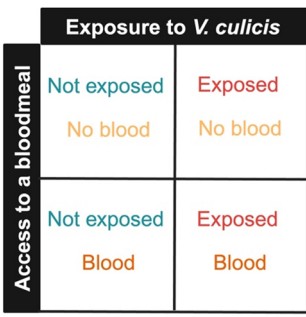
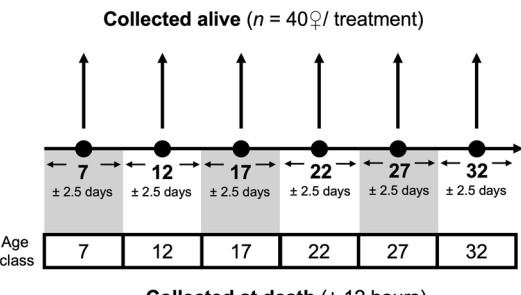

**Fig. 2 | Effect of infection and blood meal on longevity and spore load. a** Age at death for individuals exposed or not with *V. culicis*, that took or not a blood meal. Individuals who were collected alive were excluded from this analysis. The sample sizes (*n*) for each treatment are shown above the data points. **b** Individual spore load at death for individuals that had a bloodmeal (orange) or not (yellow). A smooth line was drawn for ease of visualisation with the "loess" method, and the sample sizes were 194 for females who had a blood meal and 199 if they did not. The error bars show the standard error of the mean (SEM), and the letters indicate statistically significant differences from multiple comparisons.

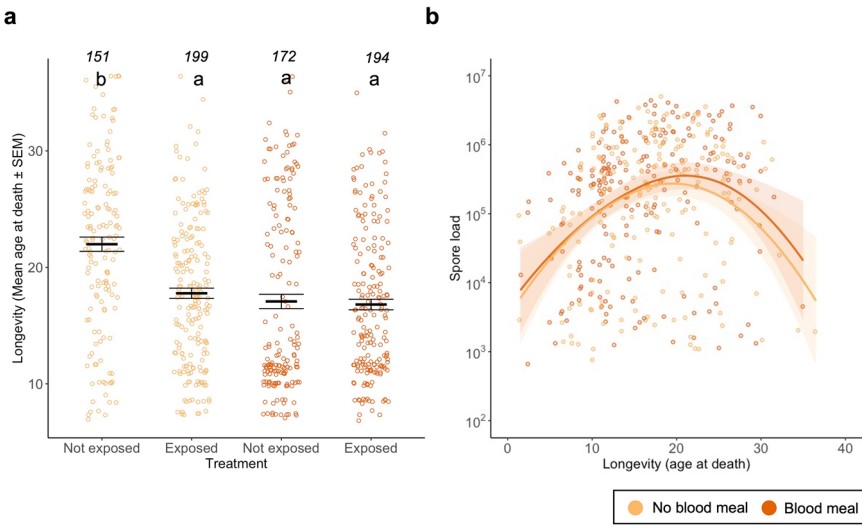

understanding of the evolution of vector- and non-vector-borne diseases. Lastly, it emphasises the need for closer integration of ecology and epidemiology in vector biology research.

## Results

### Effect of infection and blood meal on individuals that died naturally

To begin, we collected individuals that died naturally every 12 hours (Fig. 1) and evaluated how infection during the larval stage and blood-feeding on day 7 of adulthood affected mosquito longevity and parasite development.

Alive mosquitoes lived, on average, about 18.4 days. While neither exposure to *V. culicis* ($\chi^2 = 0.13$, df = 1, $p = 0.716$) nor blood-feeding ($\chi^2 = 1.87$, df = 1, p = 0.171) had a main effect on longevity, the exposure reduced the longevity of unfed mosquitoes by 4.2 days (interaction exposure * blood meal: $\chi^2 = 14.13$, df = 1, $p < 0.001$, Fig. 2a). Exposed mosquitoes harboured, on average, about $6.6 \times 10^5$ spores, and bloodfed individuals had slightly higher parasite load ($\chi^2 = 3.91$, df = 1, $p = 0.047$). However, spore load increased and then decreased as age at death increased ($\chi^2 = 12.96$, df = 1, $p < 0.001$), particularly in unfed individuals (interaction age at death * blood meal: $\chi^2 = 5.13$, df = 1, $p = 0.023$, Fig. 2b).

Altogether, both infection with *V. culicis* or blood-feeding reduced mosquito lifespan to a similar extent. However, being exposed to both stressors did not lead to greater mortality than either stressor alone, with individuals living on average 17 days.

### Spore load and resource dynamics in alive and dead mosquitoes across age classes.

Next, we investigated spore load and energy resource dynamics in both alive and dead mosquitoes. Alive individuals were sampled every 5 days from day 7 onward, while dead individuals were collected every 12 hours and grouped into age classes to match those of alive samples (Fig. 1).

Spore load (Fig. 3a) increased with the age of mosquitoes ($\chi^2 = 27.59$, df = 5, $p < 0.001$) and was higher in dead than alive mosquitoes (dead: $5.0 \times 10^5$; alive: $2.1 \times 10^5$; $\chi^2 = 8.32$, df = 1, $p = 0.004$, Supplementary Fig S1), independently of their age (interaction death status * age: $\chi^2 = 3.64$, df = 5, $p = 0.602$). On average, blood-fed mosquitoes harboured similar spore loads than unfed individuals ($\chi^2 = 1.25$, df = 1, $p = 0.263$) except for day 22 and 27 (interaction blood * age: $\chi^2 = 14.18$, df = 5, $p = 0.014$). These effects were independent of any other combination of the treatments (for further details on the statistical analysis, see Supplementary Table S2).

Protein content (Fig. 3b) was highest at age 7, lowest at ages 12 and 17, and intermediate at ages 22, 27 and 32 ($\chi^2 = 17.31$, df = 5, $p = 0.004$), and was higher in dead than alive mosquitoes ($\chi^2 = 15.15$, df = 1, $p < 0.001$) although this effect was dependent of the age of the mosquitoes (interaction death status * age: $\chi^2 = 26.32$, df = 5, $p < 0.001$; Fig. 3b). It was similar in blood-fed and unfed mosquitoes ($\chi^2 = 0.06$, df = 1, $p = 0.799$), and was independent of the infection status ($\chi^2 = 0.47$, df = 1, $p = 0.492$) or any other combination of the treatments (for further details on the statistical analysis see Supplementary Table S2).

Carbohydrate content (Fig. 3c) decreased from age 7 to age 22 and then increased as mosquitoes aged further ($\chi^2 = 50.57$, df = 5, $p < 0.001$). Individuals that died naturally had fewer carbohydrates than alive ones that we sacrificed ($\chi^2 = 7.02$, df = 1, $p = 0.008$), and this difference was stronger in the youngest and oldest mosquitoes than at other ages (interaction death status * age: $\chi^2 = 34.21$, df = 5, $p < 0.001$; Fig. 4c). It was similar in blood-fed and unfed mosquitoes ($\chi^2 = 0.05$, df = 1, $p = 0.820$), and was independent of the infection status ($\chi^2 = 0.01$, df = 1, $p = 0.910$) or any other combination of the treatments (for further details on the statistical analysis see Supplementary Table S2).

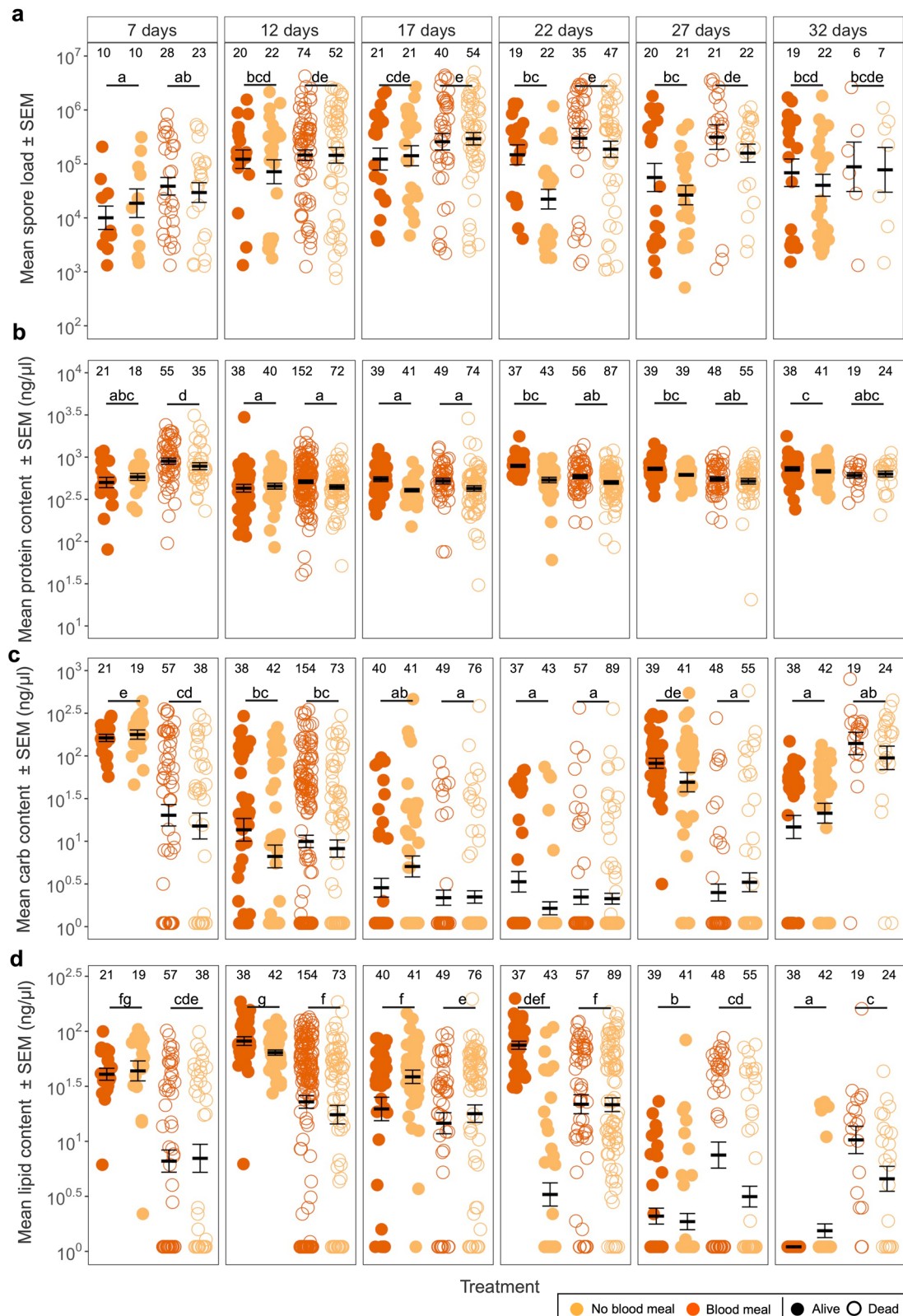

**Fig. 3 | Spore and resource dynamics. a** Individual spore load, **b** protein, **c** carbohydrate and **d** lipid content throughout their lifetime for individuals that took a blood meal, or not, and were collected alive or dead. Individuals collected at death were allocated into subgroups (e.g., individuals that died between days 4.5 and 9.5 were assigned to age class 7, while those that died between days 9.5 and 14.5 were assigned to age class 12, and so on) so that they could be compared to the individuals collected alive (for more details see Fig. 1). The error bars show the SEM, and the letters indicate statistically significant differences from multiple comparisons between alive and dead individuals. The sample sizes (*n*) for each treatment and comparison can be found above the respective treatment of each plot. Panel assembled in Biorender.com.

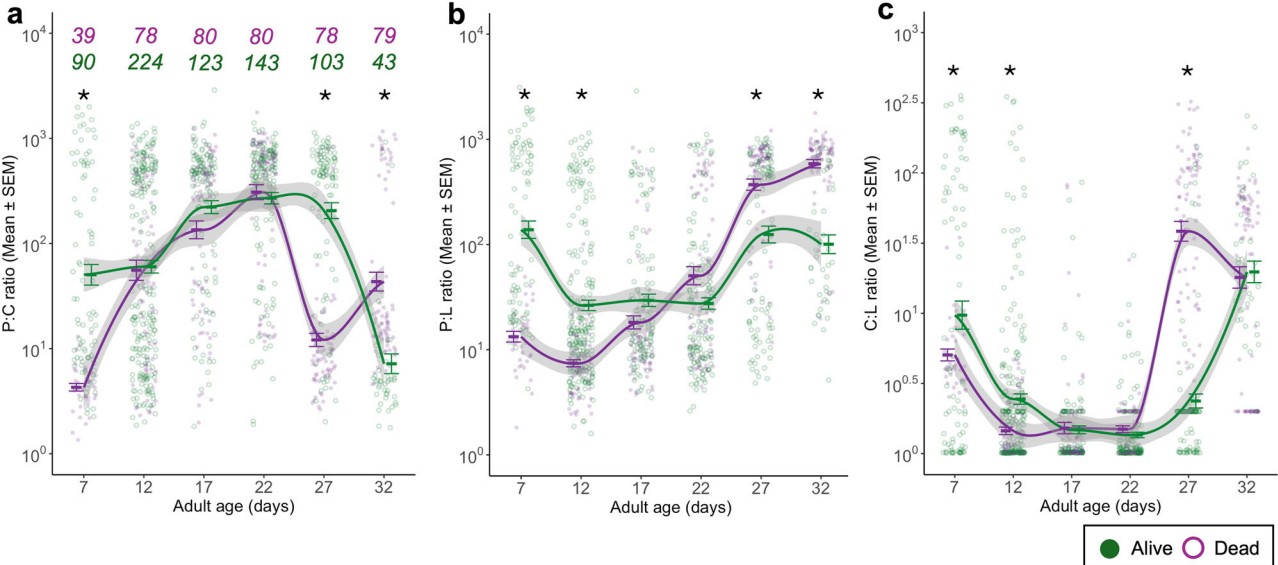

**Fig. 4 | Resource ratios.** Individual protein:carbohydrate (P:C) (**a**), protein:lipid (P:L) (**b**) and carbohydrate:lipid (C:L) (**c**) ratios throughout their lifetime for individuals that were collected alive (green) or dead (purple). Individuals collected at death were allocated into subgroups (e.g., individuals that died between days 4.5 and 9.5 were assigned to age class 7, while those that died between days 9.5 and 14.5 were assigned to age class 12, and so on) so that they could be compared to the individuals collected alive (for more details see Fig. 1). As carbohydrates and lipids are energetic sources, their ratio is inherently energetic. The sample sizes (*n*) are the same across the different ratios and are shown above the plot (**a**), indicated with the corresponding treatment colour. The error bars show the SEM, and the asterisk (*) indicates statistically significant differences between dead and alive individuals for that time point.

Lipid content (Fig. 3d) increased from age 7 to 12 and then decreased to the lowest value in the oldest individuals ($\chi^2 = 52.16$, df = 5, $p < 0.001$). On average, individuals that died naturally had similar levels of lipids than those that were alive ($\chi^2 = 0.52$, df = 1, $p = 0.472$), but their content was highly dependent of their age and death status, as lipid content decreased with age in alive mosquitoes while it remained constant in dead mosquitoes (interaction death status * age: $\chi^2 = 24.18$, df = 5, $p < 0.001$; Fig. 3d). It was similar in blood-fed and unfed mosquitoes ($\chi^2 = 0.01$, df = 1, $p = 0.907$), and was independent of the infection status ($\chi^2 = 0.73$, df = 1, $p = 0.391$) or any other combination of the treatments (for further details on the statistical analysis see Supplementary Table S2).

Altogether, spore load and energy resource levels varied with mosquito age, but dead individuals consistently harboured more spores and had lower carbohydrate levels. More importantly, alive and dead mosquitoes showed distinct trajectories in protein, carbohydrate, and lipid dynamics, reinforcing that physiological deterioration and energy imbalance are associated with mortality risk.

**Resource ratios dynamics in alive and dead mosquitoes across age classes.** Finally, using the same set of individuals (Fig. 1), we analysed the dynamics of resource ratios (protein-to-carbohydrate, protein-to-lipid, and carbohydrate-to-lipid) in alive and dead mosquitoes to further characterize the metabolic shifts associated with survival likelihood observed in the previous section.

The ratio of proteins to carbohydrates increased until age 22 and then decreased ($\chi^2 = 199.24$, df = 5, $p < 0.001$). Overall, alive mosquitoes had lower P:C ratio than the ones that died naturally ($\chi^2 = 96.36$, df = 1, $p < 0.001$), but this difference was most apparent for the youngest mosquitoes and at day 27 whereas at age 32, alive mosquitoes had a slightly higher P:C ratio (interaction death status * age: $\chi^2 = 123.75$, df = 5, $p < 0.001$; Fig. 4a). For further details on the statistical analysis see Supplementary Table S3.

The ratio of proteins to lipids was highest at ages 7, 27 and 32, and lowest at ages 17 and 22 ($\chi^2 = 191.11$, df = 5, $p < 0.001$). Up to age 17, alive mosquitoes had a lower P:L ratio those that died naturally, but older mosquitoes showed the opposite pattern (interaction death status * age: $\chi^2 = 104.19$, df = 5, $p < 0.001$; Fig. 4b) leading to alive mosquitoes having overall higher P:L ratios ($\chi^2 = 40.96$, df = 1, $p < 0.001$). For further details on the statistical analysis, see Supplementary Table S3.

The ratio of carbohydrates to lipids was highest at ages 7, 27 and 32, and lowest at ages 17 and 22 ($\chi^2 = 153.87$, df = 5, $p < 0.001$). Overall, this ratio was lower in alive mosquitoes than in those that died naturally ($\chi^2 = 10.97$, df = 1, $p < 0.001$), particularly if they were younger than 17 days old, but higher, if they were older (interaction death status * age: $\chi^2 = 58.72$, df = 5, $p < 0.001$; Fig. 4c). For further details on the statistical analysis see Supplementary Table S3.

Altogether, shifts in protein-to-carbohydrate, protein-to-lipid, and carbohydrate-to-lipid ratios consistently differed between alive and dead mosquitoes across age classes. These patterns suggest that the ability to regulate energy allocation—particularly the shift in carbohydrate-to-lipid ratio in older individuals—is closely associated with survival outcomes.

## Discussion

Our findings show complex relationships between ecological factors that the main malaria vector, *An. gambiae*, is commonly exposed to (blood meal and parasitism) and longevity, which is a fundamental life history trait determining the mosquito's fitness and vectorial capacity. These results also suggest that shifts in resource allocation may underlie the observed patterns, particularly the effects of blood-feeding and microsporidian infection on survival, and the tight link between resource shifts and lifespan. Notably, although the resource levels throughout the adult mosquito's life were not influenced by blood meal or by infection with a microsporidian, they were closely linked to survivorship.

Taking a blood meal shortened the mosquitoes' lives (Fig. 2a), most likely due to increased oxidative stress and gut tissue damage in mosquitoes[13,14,26,27]. In our design, individuals were allowed to fully engorge on blood during the feeding period, potentially exacerbating these costs, particularly concerning tissue damage. Infection by *V. culicis* also shortened the lifespan, corroborating other studies[23,24,28]. The infection cost appeared to be linked to the spore load, for mosquitoes that died naturally harboured more spores than those alive of the same age. More surprisingly, there was a cost of infection only in mosquitoes that had not blood-fed, although blood-feeding increased the spore load. A possible explanation is that both the

production of spores and the host's survival are limited by resources replenished by blood-feeding.

Our findings highlight that there are clear shifts in the ratios of resources during the life of the mosquitoes. As protein content remained relatively stable over the lifespan (Fig. 3b), the shifts of P:C and P:L predominantly stem from differences in carbohydrate and lipid levels, respectively (Fig. 4ab). In particular, individuals at death consistently exhibited elevated P:C and P:L ratios over time (Fig. 4ab), indicative of reduced carbohydrate and lipid reserves. Conversely, alive individuals show dynamic energy use, primarily relying on carbohydrates until day 22, after which lipid reserves became the predominant fuel source (Fig. 4c). Below, we propose a few complementary explanations for these findings.

Lipid reserves are a last resort energy source, less accessible to the host than carbohydrates, prompting individuals to primarily rely on carbohydrates. Given that mosquitoes had continuous access to a 6% sucrose solution, we expected them to primarily use carbohydrates for energy until their reserves were depleted or access to the sucrose became impaired. Around day 20, we observed that some individuals began to struggle to reach the top of the cup where the sucrose cotton balls were placed, possibly due to reduced mobility associated with aging. This impaired access may have limited their ability to replenish carbohydrate stores, contributing to a net decrease in carbohydrate content and triggering a metabolic shift toward lipid utilisation. Interestingly, this energetic shift may coincide with the timing when a new cluster of eggs is typically produced. It has been proposed that in several anautogenous arthropods, including mosquitoes such as *Aedes aegypti* and *Anopheles gambiae*, the initiation of vitellogenesis may rely not only on energetic resources but also on iron signalling triggered by blood ingestion[9,29–31]. If such is the case for *An. gambiae*, individuals likely begin investing in a new egg batch shortly after laying the first, around day 12, resulting in increased carbohydrate consumption. This is supported by a marked decrease in carbohydrate content from day 12 to day 22 in alive individuals (Fig. 4ac). Lipids are initially used to support homoeostasis and later become the primary energy source. After completing the energetic demands of reproduction, surviving mosquitoes appear to replenish carbohydrate reserves using the available sucrose, as reflected in the increased levels observed from day 27 to day 32 (Fig. 4bc).

Moreover, when comparing the overall energetic use between individuals collected alive *versus* those collected at death over the same period, distinct energetic trajectories emerge. In particular, individuals that die exhibit lower energetic profiles and, as early as seven days post-emergence, tend to have fewer energetic reserves compared to their surviving counterparts. This observation suggests that energetic reserves measured early in adult life may hold predictive value not only for resource acquisition but also for longevity. Constantly lower energetic reserves likely correspond to reduced physical condition and overall fitness, increasing susceptibility to mortality. Furthermore, personal observations indicate that mosquitoes nearing death exhibit reduced activity and tend to remain at the bottom of the cup. In such compromised conditions, accessing sucrose at the cup's surface becomes challenging, hindering resource replenishment and increasing both the usage of lipids reserves and the mortality risk. Conversely, individuals with higher baseline energetic reserves may retain the energy to fly and access available sucrose, as evidenced by carbohydrate replenishment observed on day 27 in surviving mosquitoes (Fig. 4c).

Thus, we propose that energetic reserves at adulthood's onset significantly influence and predict longevity. Consistent with the findings of Bedhomme *et al.* (2004) in *Ae. aegypti* larvae, which showed significantly reduced lipid, sugar, and glycogen levels in *V. culicis*-infected individuals compared to uninfected ones[32], our results extend these insights to adults stage and to a different species, *An. gambiae*. Although larval-stage deficits may appear modest, our data reveal the cumulative energetic toll of infection over the mosquito's lifespan. By day 7, infected mosquitoes display carbohydrate levels similar to uninfected individuals (189 vs. 187 ng/µL) but retain only half the lipid levels (38 vs. 61 ng/µL, Supplementary Fig S3). This underscores the long-term energetic costs of microsporidian infections on lipid reserves, while also demonstrating that infected mosquitoes can recover carbohydrate levels by feeding on sucrose, compensating for larval-stage deficits and achieving parity with uninfected counterparts.

Through a vector biology perspective, our findings reveal a noteworthy pattern with significant implications for epidemiology. As mentioned above, *An. gambiae* is known as the main *Plasmodium* vector. A study by Costa and colleagues (2018) demonstrated that the within-mosquito resource environment affects *Plasmodium* infection success and virulence, in particular regarding lipid exploitation. For instance, only 20% of the mice bitten by mosquitoes with lipid depletion were infected with *Plasmodium* compared to the ones bitten by control mosquitoes, which had a 90% infection success[33]. This result was attributed to the need for lipids for the correct development of *Plasmodium* sporozoites and oocysts. Lipids are not an easy resource for parasites to access: structural lipids can be complicated for parasites to access, and hence, many parasites, such as *Plasmodium*, tend to focus on circulating or more available forms of lipids. These can include lipids being transported in specialised transporters[34,35]; lipid droplets circulating in the cytoplasm of a wide variety of cells[36]; in general, they are heavily stored in the fat body and moved to other organs (such as the ovaries) as needed[34,37]. Costa and colleagues then proposed that the duration of *Plasmodium* sporogony, typically occurring 13 to 16 days after a blood meal, is an evolutionary adaptation to the temporal dynamics of mosquito reproduction. Our results seem to lend support to this hypothesis as we observe that the major shifts in lipid reserve mobility align with the expected timing of sporogony in *Plasmodium*-infected mosquitoes, suggesting an evolutionary adaptation of this parasite to timely exploit mosquito resource dynamics.

Altogether, our study highlights the significance of accounting for resource availability and usage in mosquito life studies. Our findings indicate that resource reserves are affected by the individual life history traits and environmental stressors, such as parasites. We demonstrated that blood meals impact mosquito longevity and resource dynamics, which might contribute to changes in energetic shifts within the mosquito. Furthermore, our evidence also suggests that the quantity and dynamics of energetic reserves may serve as predictive factors for mosquito survival. We believe that these results will contribute to a better understanding and prediction of the evolution of *Plasmodium* infections and hope to inspire similar studies in other vector and non-vector infection models.

## Limitations of the study

In this study, we demonstrated the predictive and deterministic role of energy reserves throughout the life history of *Anopheles gambiae*. However, one major limitation of our work is the lack of experimental testing on resource dynamics during *Plasmodium* infection and co-infection with *V. culicis*, which is crucial to evaluate our hypothesis regarding resource competition during co-infection.

Another limitation is that our initial measurements were taken on day 7. While this timing ensured that all mosquitoes harboured spores, allowing us to quantify spore load, it also gave them time to replenish their resources, particularly carbohydrates, by feeding on the sucrose source, potentially hindering the true impact of infection on some measured reserves. Measuring energy reserves immediately after adult emergence would provide a clearer baseline of resources before feeding begins. Additionally, assessing resource dynamics at critical life stages (such as immediately before pupation, during the pupal stage, and just after emergence) would offer valuable insights into the energetic costs of metamorphosis. This approach would help quantify how these transformative stages deplete resources and how infection amplifies these costs, shedding light on the trade-offs imposed by infection during key developmental transitions.

Finally, our design included only a single blood-feeding event, with resources measured at 12 days post-eclosion. Although this time point was chosen to allow sufficient digestion and allocation of blood-derived nutrients, it prevents us from resolving the temporal dynamics of post-blood meal metabolism. Moreover, restricting females to one blood meal does not reflect the repeated feeding cycles typical in nature. These choices were

necessary to maintain a full factorial design across multiple treatments and to ensure adequate sample sizes, but they limit the temporal and ecological resolution of our findings.

## Materials and Methods

**Experimental system.** The host of our experiment was the Kisumu strain of *Anopheles gambiae s.s*[38], which we had maintained at a density of about 600 individuals per cage and standard lab conditions (26 ± 1°C, 70 ± 5% relative humidity and 12 h light/dark) for several years before the experiments. During this time, cohorts were initiated weekly and kept in separate cages throughout their lives, and eggs were obtained from several cages. Our parasite was the microsporidian *Vavraia culicis floridensis* (provided by James Becnel, USDA, Gainesville, FL), which is a generalist, obligatory, intracellular parasite that infects mosquito epithelial gut cells[24]. Before beginning the study, we maintained *V. culicis* in large numbers by alternatingly infecting *Aedes aegypti* and *An. gambiae* populations to ensure that it remains a generalist parasite.

**Experimental design.** To start the main experiment, we combined eggs from three cages, which held mosquitoes of different ages. We used a cross-factorial design, where one factor was exposure or not to *V. culicis*, and the other was blood-fed or not. Freshly hatched *An. gambiae* larvae were moved individually into the wells of 12-well culture plates, each containing 3 ml of distilled water. They were fed daily with Tetramin Baby ® fish food according to their age: 0.04, 0.06, 0.08, 0.16, 0.32 and 0.6 mg/larva at ages 0, 1, 2, 3, 4 and 5 days or older, respectively[39,40]. Two-day-old larvae were exposed to 0 or 10,000 spores of *V. culicis*. Pupae were moved to sets of three (21 ×21 x 21 cm) cages per treatment group, each containing approximately 200 individuals. Mosquitoes emerged these cages and had constant access to a 6% sucrose solution. The males were removed upon emergence, and untreated males from the colony were added to the cages at a number equal to the number of females, who were left to mate for seven days.

Seven days later, the males used for mating were removed. Female mosquitoes were offered a single blood meal on human skin (TGZ) for five minutes, and only fully engorged individuals were retained in the experiment to ensure consistent feeding status. Two days later, the remaining mosquitoes were moved individually to 150 ml cups (5 cm Ø x 10 cm) that were covered with a net and contained a 50 mm Ø petri dish (to keep the mosquitoes from drowning) on the surface of 50 ml deionised water, a 10 × 7 cm rectangular filter paper (to keep the air humid) and a 110 mm Ø Whatman filter paper. They remained in these cups for 48 hours to allow egg laying. Although fecundity data were collected, they are not presented because they are not relevant to this study. This is because the fecundity cost of *V. culicis* infection in *An. gambiae*, using the same infection protocol and dose, has already been extensively described in the literature, and our focus in this manuscript was instead on resource dynamics. After this period, the same individuals were transferred to new 150 ml cups, each covered with a net and containing a 50 mm Ø petri dish floating on 50 ml of deionized water (to prevent drowning) and a 10 × 7 cm piece of filter paper to maintain humidity. These mosquitoes were then monitored for the longevity assay and continued to receive 6% sucrose solution throughout both assays.

To facilitate comparisons between alive and dead individuals, we assigned an age class to mosquitoes based on their collection day. Alive individuals were collected at five-day intervals starting on day 7 (before the bloodmeal) and continuing through day 32. Dead individuals were collected every 12 hours as they naturally died. The 12-hour period was based on preliminary results (further details in Supplementary Fig S2) that showed no significant degradation of any of the resources for at least 18 hours after death. The collected mosquitoes were frozen at – 80°C until we assayed spores, proteins, carbohydrates and lipids.

Dead mosquitoes were assigned to the same age class as their closest alive counterparts to ensure meaningful comparisons between individuals collected alive and dead. Specifically, a mosquito that died within ±2.5 days of an alive collection day was grouped into the corresponding age class

(Fig. 1). For example, individuals that died between days 9.5 and 14.5 were assigned to age class 12, and so on. This approach ensured that dead mosquitoes were temporally proximate to the alive individuals in the same age class, thereby improving the comparability of the two groups. The only exception was mosquitoes that died before day 7, which were excluded from the analysis because the blood-meal treatment was administered on day 7. Thus, the mosquitoes dying before this point had not received their assigned treatment.

**Spore load.** Since spore load and resource content are directly related to body size, all individuals were weighed and transferred to a 1.5 ml microcentrifuge tube. Then, 100 µl of extraction buffer (100 mM $KH_2PO_4$; 1 mM dithiothreitol; 1 mM EDTA) per mg of the mosquito was added to each tube, and the mosquitoes were homogenised at 30 Hz for three minutes with a stainless-steel bead (Ø 5 mm) and a Qiagen TissueLyser LT. From the homogenate, 20 µl were used to measure the spore load in a haemocytometer with a phase-contrast microscope (400x magnification). Spore load per individual mosquito was normalised based on their body size. All of the mosquitoes exposed to the parasite were infected.

**Resources.** The remaining homogenate solution was used to quantify total protein, total carbohydrate and total lipid content through colourimetry with a SpectraMax i3x plate reader[41]. For each plate (and therefore resource), an eight-multipoint standard curve was prepared with 10 to 400 µg/mL of albumin (for proteins), 1 to 40 µg/mL of glucose (for carbohydrates) or 5 to 200 µg/mL of triolein (for lipids). All samples and points of the standard curve were measured in duplicate to account for experimental variation.

Proteins were measured with the Bradford method[42], which consists of centrifuging the samples at 12,000 rpm at 4 °C for four minutes, transferring 5 µl of the supernatant into a 96-well plate, adding 200 µl of Bradford reagent (Sigma-Aldrich) to the samples and standards, and incubating them at room temperature with soft agitation for 15 minutes. Optical density (OD) was then read at 595 nm.

For carbohydrates quantification, we added 20 µl of 20% sodium sulphate, 5 µl of extraction buffer and 1500 µL of chloroform-methanol solution (1:2) to the remaining homogenate solution, briefly mixed the sample, and centrifuged it at 12,000 rpm at 4 °C for 15 minutes. 300 µl of the supernatant was placed into a quartz 96-well plate (Hellma) and left to evaporate for 40 minutes under the hood. 240 µl of 1.42 g/l anthrone reagent was added to each sample and each standard. The solutions were incubated for 15 minutes at room temperature and a further 15 minutes in a water bath at 90 °C. The optical density was then read at 625 nm.

Lipid content was assessed by pipetting 200 µl of the remaining solution into a quartz microplate and incubating the plate in a preheated oven at 90 °C for about 30 minutes. When all the solvent had evaporated, 10 µl of 98% sulphuric acid was added to each well and incubated in a water bath at 90 °C for two minutes. After ice-cooling the plate for a few minutes, 190 µl of vanillin reagent was added to each well, and the plate was sealed and incubated for 15 minutes with gentle agitation. Optical density was read at 525 nm. All the resources were quantified according to the standard curves with the software SoftMax Pro version 7.1 in the SpectraMax i3x plate reader.

It is noteworthy to say that resource quantification was performed at 12 days post-adult emergence to allow sufficient time for blood meal digestion and for blood-derived resources to be metabolised, stored, or utilised. While circulating nutrients were no longer directly measurable at this later stage, stored resources (particularly lipids and carbohydrates) provided a more stable indicator of the physiological impact of blood-feeding. Proteins, which are more rapidly allocated to reproduction and metabolism, were expected to be more variable at this time point.

**Statistics and reproducibility.** All analyses were conducted with R version 4.4.2, using the packages "DHARMa"[43], "car"[44], "glmmTMB"[45], "emmeans"[46] and "multcomp"[47], Significance was assessed with the

Anova function of the "car" package[44] We used a type III ANOVA in the case of a significant interaction and a type II ANOVA otherwise. When relevant, we performed post-hoc multiple comparisons with the package emmeans, using the default Tukey adjustment. The data and R script for the analyses described below are available as Supplementary Data.

We initially tested for the effects of blood-feeding, infection status, and survival status on the resource ratio. As blood-feeding and infection status showed no significant or only marginal effects, we present only the effect of survival status in the main text. Complete model outputs, including all tested interactions, are provided in the Supplementary Table S1.

Age at death of the mosquitoes that died naturally was analysed with a linear model with a Gaussian distribution of errors, where the response variable was age at death in days and the explanatory variables were infection status, blood meal and their interactions.

Mosquitoes with detectable spores were then used to assess the spore load. Because we counted the number of spores in a haemocytometer containing 0.1 μl of the sample (i.e. 1/1000 of the total volume), the detection threshold was estimated to be 1000 spores. Spore load at death (from mosquitoes that died naturally) was analysed using a generalised linear model with a negative binomial distribution of errors[48] where the response variable was spore load and the explanatory variables were age at death, blood meal and their interactions.

We then tested whether the spore load differed between blood-fed and unfed mosquitoes and between those that died naturally or were collected alive. We first pooled the mosquitoes that died within 2.5 days of an age at collection (i.e., 7, 12, 17, 22, 27 and 32 days) so that the time points for the mosquitoes that died naturally or that were alive were the same. Since spores do not appear before about four days after emergence and since very few mosquitoes lived beyond 34 days, we omitted the youngest and oldest individuals. We analysed the spore load of the mosquitoes that were infected with a generalised linear model with a negative binomial distribution of errors, where the explanatory variables were blood meal, death status, age at collection and their interactions.

To analyze the mosquitoes' protein, carbohydrate, and lipid content, we again pooled the dead mosquitoes into groups corresponding to the ages at which the living mosquitoes were collected. Because of the distribution of the resource contents, protein content was analysed with a generalised linear model with a Gamma distribution of errors, whereas carbohydrates and lipids were analysed with a generalised linear model with a Tweedie distribution of errors. The explanatory variables were blood meal, death status, infection by *V. culicis*, age at collection and their interactions. See Supplementary Table S2.

Finally, we assessed differences in the ratios between resources, i.e., protein:carbohydrate (P:C), protein: lipid (P:L), and carbohydrate:lipid (C:L) ratio, as a proxy for energetic shifts in the host. Since some of the resource levels were 0, we added 1 to each value before calculating the ratio. The ratios were analysed with a generalised linear model with a Gamma distribution of errors, where the explanatory variables were blood meal, death status, infection by *V. culicis*, age at collection and their interactions.

## Reporting summary
Further information on research design is available in the Nature Portfolio Reporting Summary linked to this article.

## Data availability
The data used in this study, along with the R script containing the code for the statistical analyses and figures presented in this manuscript, are available as Supplementary Data.

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

## Acknowledgements
We thank Alfonso Rojas Mora for the technical advice. We also thank Tadeusz J. Kawecki for feedback and constructive comments on the manuscript. TGZ, LMS, and the project were supported by the SNF grant 310030_192786.

## Author contributions
L.M.S., J.C.K. and T.G.Z. conceived and designed the experiments. L.M.S. and T.G.Z. collected, analysed and wrote the first draft of the manuscript. All authors contributed critically to the drafts.

## Competing interests
The authors declare no competing interests.
