## [Transparent Peer Review file · Communications Biology]

Energetic shifts reflect survival likelihood in *Anopheles gambiae*

Corresponding Author: Dr Luis Silva

Version 0:

Reviewer comments:

Reviewer #1

(Remarks to the Author)

In this study, Zeferino et al. examined how blood meal and infection by *Vavraia culicis* affect mosquito (*An. gambiae*) longevity and resource kinetics. They found that the interaction of both stressors has a cost on lifespan. Interestingly, surviving mosquitoes efficiently shift from using carbohydrates to lipids for energy, a trait missing in those that die naturally. This might suggest that energy metabolism plays a key role in survival, highlighting the intricate evolutionary relationship between hosts and parasites.

Strengths: The authors encompassed in their analysis a large range of adult mosquito lifespan, and analysed also dead individuals, which added novelty in the study. Moreover, the experimental design is well controlled.

Limitation: As the authors disclose themselves, it would be interesting to explore if these findings hold also when the blood meal is infected with malaria parasites.

General comments

1) Title and abstract

The authors discuss the hypothesis that the energetic shift observed in dead versus collected alive mosquitoes might be a predictive parameter for mortality/longevity (lines 60-61, and 192-202). However, the title states "Energetic shifts predict the mortality of *Anopheles gambiae*", and in the abstract we read "shifts in carbohydrate-to-lipid ratios predict survival likelihood". In my understanding, the word "predict" implicates to obtain specific predictive values through modeling work, with the probable need to use data from independent experiments, broken down in training and testing datasets. Moreover, experimentally it would be interesting to analyze not dead but close-to-death mosquitoes (cited by the authors at lines 195-196) for predictive purposes, as one cannot exclude the drop in changes levels happens immediately after death. The data analyzed in this manuscript show association between survival status and energetic shifts, however in my understanding no modeling work was done to clarify prediction values of the different resource readouts. Although it is interesting to discuss this hypothesis, in the absence of further mathematical analysis, the word "predict" in the title (line 1) and in the abstract (line 18) should be replaced "are associated with" or similar, to align the claim with the data shown.

Lines 20-22. The authors put their results in the context of the current knowledge about *Plasmodium* dynamics in the *Anopheles* host. However, no *Plasmodium* experiments were carried out in the present manuscript. I think these are interesting hypotheses which deserve to be examined in the discussion section, but misleading to be reported on the abstract. I suggest to remove this sentence and to stick to introduction-rational-method-results-conclusion structure in the abstract.

2) Results section

I have a couple of suggestions to improve readability of the results section. The section starts directly into the results details. I think it would help the reader to quickly recap on the main aim and experiment design before that. In particular, I don't mean the details on how the experiment was done (which is extensively found in the methods) but summarize the choices of infection time (during larval development), blood feeding time, egg laying, and time points of collections for both alive and

dead mosquitoes, related to the questions asked. This would provide a more straightforward understanding of the figures and result sections. Moreover, for the same reasons, I also suggest to add a conclusion paragraph to each result subsection, to help delineating the major findings, before they are discussed in details in the following section.

3) Figures

The authors can consider indicating the sample sizes directly in the graphs, below the corresponding dotplot column, for Fig. 1A and Fig. 2. It would lighten and increase readability of the figure legends.

For Fig. 2 and 3, alive and dead individuals are compared side by side; however, while live samples correspond to the indicated time point in the axis, dead individuals were pooled for ± 2.5 around the indicated time point (as explained in details in the methods, line 314). I think it's important to mention this point also in the corresponding figure legend.

In Fig. 3, I suggest to choose two colors that differ more between each other, for the "dead" vs "alive" comparisons; I personally find the graph difficult to read with the current colors.

Specific comments:

4) Line 52. Citation 19 refers to another mosquito, *A. aegypti*; please make it clear in the text to avoid confusions.

5) Lines 58-59. The authors state "Our findings show that the resource levels throughout the adult mosquito's life are influenced by blood meal and infection". Moreover, the authors state at lines 154-156 "In particular, the resource levels throughout the adult mosquito's life were influenced by blood meal and by the infection by a microsporidian. In addition, at lines 207-209, it reads "By adulthood, infected mosquitoes exhibit resource depletion of 2-3 times higher than their uninfected counterparts (e.g., carbohydrates: 64 vs. 187 ng/ μ L; lipids: 20 vs. 48 ng/ μ L by day 7). However, in my understanding, resource content was similar in blood-fed and unfed mosquitoes and was independent of the infection status, or combination of the treatments (Table S2), as also mentioned in the results (lines 95-98, 102-105, and 110-113). I might have misunderstood some point, but this seems in apparent contradiction to me. Please clarify / harmonize throughout the text.

6) Line 60. The authors state "resources were tightly linked to adult survivorship". I am not sure I understand this sentence: (some) resource levels? (specific) resource shifts? Please specify.

7) Line 61. The word "understanding" is repeated twice in the same sentence, please rephrase for clarity.

8) Lines 76-77. Since collected alive individuals were excluded from this graph (Fig. A), please remove the "alive" "dead" legend in the right lower corner of the figure, as it makes the reader expecting to see also live mosquitoes plotted in the graphs.

9) Lines 85-87. The authors state "Spore load ... was higher in dead than alive mosquitoes, independently of their age". This concept is also taken back in the discussion at lines 160-162. Although the statistical significance of this result is reported in the results and shown in Table S2, it could help to show an additional graph with spore load in dead vs alive mosquitoes all timepoints merged, for the reader to visually appreciate cumulative effect size.

10) Lines 116-117. Please clarify between which groups the multiple comparisons were applied. It is not clear to me because the strait line in the graphs spans over two groups.

11) Line 153-154. The authors state "They suggest the role of resource allocation underlying these relationship." I am not sure I understand. Do they mean "They suggest that resource allocation plays a role underlying these relationships"? Please clarify.

12) Line 157. The author states "Taking a blood meal strongly shortened the mosquitoes' lives". However, accordingly to Figure 1a the reduction corresponds only to about 3.3 days on a mean lifespan of 18, and is true only when comparing within uninfected mosquitoes. "Strongly shortened" seems overstating to me, please rephrase.

13) Lines 158-159. The authors state "In our design, individuals had ad libitum access to blood", however in the methods is reported that single mosquitoes were left to feed one time for 5 minutes (line 254). I think "ad libitum" is misleading, as it's normally intended for as much or as often as desired. Please rephrase, for example by stating mosquitoes were fully engorged, or similar.

14) Lines 179-183. The authors state "Indeed, this energetic shift interestingly coincides with an expected production of a new cluster of eggs.". I haven't found though any data showing a new (second) production of eggs in the results. Please rephrase to highlight this is an hypothesis.

Moreover, the authors state "It has been discussed that several mosquito species do not need the energetic resources to produce the new cluster of eggs, but instead solely the iron signalling. However, in the citations mentioned I only found this related to ticks and kissing bugs. Please cite specifically in which mosquitoes this has been discussed or replace with (anautogenous) arthropods. Moreover, assuming that is the case for *A. gambiae* studied here, how do the authors imagine the mosquitoes activate (again) iron signaling cascade in this experimental setting in the absence of a (second) blood meal? Please revise carefully this paragraph or remove it.

15) Line 204. Citation 29 refers to *A. aegypti* larvae, please state this also in the text to clarify there is this additional differences compared to the data shown in this manuscript.

16) Line 216. The authors state "This result was shown to be due to the need for lipids by sporozoites but not oocysts.", referring to citation 30. In this reference, the authors have shown that neutral lipids are need to form virulent sporozoites and acquired during sporulation (formation of sporozoites) happening in the oocyst. In the *P. berghei* model, a mosquito lipid transporter starts to accumulate around growing oocysts from D7 onward (Fig S3). Indeed by restoring resources through an additional blood meal at D7 post infection, the sporulation can be rescued (Fig. 4). In this context the sentence cited above results imprecise, please rephrase.

17) Line 244. The authors state that each experiment "Each experiment was started with eggs from three cages". How many experiments were performed? Were all conditions repeated in all experiments? Sample sizes per each group are indicated clearly in the figure legends, but I didn't find any information about experimental replicates.

18) Line 250. How many individual mosquitoes were hosted per cage?

19) Line 254. Please replace "on TGZ's arm" with "on human skin (TGZ)", for clarity. Were mosquitoes that didn't take a blood meal excluded or used in the unfed group? Where unfed mosquitoes also moved to the egg laying cup, and later on to a normal cup again, for a mock control? Please specify.

Reviewer #2

(Remarks to the Author)

In their study 'Energetic shifts predict the mortality of *Anopheles gambiae*' the authors examine the effects of two stressors (blood feeding and infection with a microsporidian parasite) impact energy dynamics in the malaria mosquito *Anopheles gambiae*. They seek to examine how specific resources and resource ratios in this mosquito are impacted by these stressors across a range of age points, and for mosquitoes that die naturally or are killed and examined at those age points. Mosquitoes are either parasitized as larvae or not, and/or offered a chance to blood feed at 7 days post eclosion, or not. Levels of parasite spores (if infected) were quantified using a hemocytometer, while levels of proteins, carbohydrates, and lipids were quantified via colorimetric assay. The authors observe key differences in carbohydrate and lipids levels and parasite load with age, and also between mosquitoes that died naturally or were live sampled during the experiment. The findings of the study are novel and interesting and should open the way to broader investigations of interconnectivity between metabolism, immunity, and physiology in insects. However, the framing of the research is much broader than the actual experiments that were performed and there are multiple subjects in the description of the experimental design that need further clarification.

Major comments

1. The abstract and parts of the discussion mention adaptation of *Plasmodium* to mosquito host resource availability, but no *Plasmodium* model is used in this study and findings based on a single microsporidian strain are not necessarily transferrable to other, potentially more pathogen microsporidians, let alone a distinct clade of parasites like *Plasmodium*.

2. It's difficult to gauge the effects of blood feeding in this experiment. Mosquitoes were offered a single blood meal at 7 days post-eclosion. It is unclear whether they were then screened to determine feeding status (not fed, partially fed, or fully fed). Resource levels were then quantified at 7 dpe (pre-blood feeding), or at 12 dpe, which would be after most mosquitoes had finished digestion of their blood meal and blood meal-derived resources were catabolized and stored. In nature, over the course of a lifespan of approximately 30 days, it would be expected that a female mosquito would blood feed several times.

3. From the description in the methods, male and female mosquitoes were allowed time to mate prior to blood feeding. From the manuscript, it is unclear whether all female mosquitoes had mated successfully and produced eggs. It is reasonable to hypothesize that energy dynamics would differ between gravid and non-gravid females, and that this might also have been impacted by parasitism with *V. culicis*.

4. The methods sections suggests that fecundity data were collected during the experiment (mosquitoes allowed to lay eggs for 48 hours) but results of this assay and any potential fecundity costs of parasitism are not presented in the paper.

5. Other details in the methods and study design that need clarification or adjustment:

- Age of mosquitoes should be described in relation to time post-hatch for juveniles and time post-eclosion for adults, not in terms of "days old".
- There is no confirmation whether the amount of blood imbibed by parasitized mosquitoes differed from non-parasitized mosquitoes.
- It is unclear whether the experimental design was run just once or if it was repeated.
- Details on how the longevity assay was performed are not provided in the methods.
- It is unclear whether the measurement of resource quantities for each mosquito specimen were made once or taken as the average of duplicate or triplicate measurements.

6. The resource ratio data in fig. 3 appears to combine data across blood feeding and parasitism statuses. As these are key predictor variables in the study, the expectation would be that the resource ratio data be assessed for the influence of both variables. The rationale for the choice to combine the data should be addressed in the paper.

7. Longevity data are typically non-linear, as death rates in insect populations are inconstant over time. As such, it is unusual to see linear models used in their analysis. For Figure 1a, where data were collected every 12 hours, this should be an XY plot displaying the percentage survival over time rather than a dot plot. Additionally, the Y-axis depicts age at death rather than mean age at death, which warrants a change in the axis title.

Minor comments

8. The abstract and introduction highlight several limitations of study design for projects assessing energy dynamics in insects, and the final paragraph of the introduction highlights the authors "comprehensive approach" to addressing interactions between stressors, longevity, and resource levels. Given the very broad scale and scope of these interactions and what is possible to achieve in one experimental design, a more neutral choice of language would be appropriate here.

9. As the results section follows the introduction rather than the methods section, some explanation of the study design and treatments would be useful. Starting with "Mosquitoes that were not killed lived" (line 66) is a little confusing without description of the treatments. There are also inconsistencies with the naming conventions for some treatments throughout the manuscript with terms like "live", "dead" and "killed" used, which could be confusing to some readers. I suggest picking two clear treatment names and only using these terms.

10. There are lots of 100 values in the carbohydrate and lipid data. Is it possible that the lower limit of the standard curve was too high or the limit of detection/dilution factor too low?

11. Figs 1a and 1b – the key depicts filled circles for live mosquitoes, but it appears that no filled circles appear in these two plots.

12. Large portions of the Fig. 2 and 3 legends are a list of sample sizes. Is there a more effective way to provide this information, such as by providing a range of sample sizes per treatment on the figure and then providing the raw data as a supplementary file?

13. There is a disparity with information on blood feeding in the manuscript. In the methods it states that there was one five minute blood feeding opportunity, while in the discussion it states that mosquitoes had ad libitum access to blood. This should be clarified.

14. There is a disparity in how the longevity cost of *V. culicis* is described. In the introduction, it states that there is minimal effect on longevity, while in the discussion, the authors note that their findings and the findings of other studies highlight significant shortening of lifespan. This should be clarified.

15. Lines 165-166 – "Indeed, at many ages, the resource content was lower in the mosquitoes that died naturally than in those we killed" - for some ages resource content was lower but at others it was higher. The results are more complicated than made out in this statement.

16. Line 178 – "likely due to the inability to reach their sucrose source" – this needs to be explained a bit better – was the sucrose moved or were the mosquitoes less mobile with age? Is there evidence to support a change in mobility in your experiment?

17. Line 183 – "We show support for this hypothesis with a consistent decrease in" – it's not a consistent decrease if resource levels then increase again for older mosquitoes. This should be rephrased.

Version 1:

Reviewer comments:

Reviewer #1

(Remarks to the Author)

The authors have appropriately addressed my points. I think the clarity and consistency of the manuscript has improved. I have no further comments.

Reviewer #2

(Remarks to the Author)

Response to previous comments:

2. It's difficult to gauge the effects of blood feeding in this experiment. Mosquitoes were offered a single blood meal at 7 days post-eclosion. It is unclear whether they were then screened to determine feeding status (not fed, partially fed, or fully fed). Resource levels were then quantified at 7 dpe (pre-blood feeding), or at 12 dpe, which would be after most mosquitoes had finished digestion of their blood meal and blood meal-derived resources were catabolized and stored. In nature, over the course of a lifespan of approximately 30 days, it would be expected that a female mosquito would blood feed several times.

In our design, mosquitoes were offered a single blood meal at 7 days post-eclosion, and only fully

engorged individuals were retained in the experiment. This has now been clarified in the methods section to avoid confusion regarding feeding status. Regarding resource quantification, we chose 12 days post-eclosion as our primary time point to allow sufficient time for digestion and for blood meal-derived resources to be metabolized and either stored or utilized. While this means circulating nutrients from the blood meal were no longer directly measurable, we were able to assess stored metabolic resources—particularly lipids and carbohydrates—which reflect the physiological impact of blood feeding. Proteins, which may be more quickly allocated to reproductive or metabolic processes, are naturally more variable at this later time point. We agree that a more frequent sampling regime—such as daily measurements—would have provided a more accurate insight into the dynamics of resource metabolism post-blood meal. However, due to logistical constraints, such as the need to maintain a full factorial design across three experimental factors (infection status, blood feeding status, and survival status), it was not feasible to include additional time points or offer repeated blood meals. The factorial design inherently required a large number of mosquitoes, and ensuring sufficient sample sizes—especially for individuals that died naturally—necessitated prioritizing critical time points over high resolution. Adding extra blood meals or spreading data collection across different experimental batches would have further complicated interpretation, particularly given the sensitivity of metabolic measurements to environmental and temporal variation. We hope this explanation clarifies the rationale and experimental constraints that shaped our design decisions and the organization of the study.

Follow up: I agree with what you have said in your response that it would have been impractical to repeat the metabolite quantification for additional time points. To clarify the choice of a 12 dpe time point, it would be good to add the rationale you provided in your rebuttal to the methods section. Additionally, my original point should be addressed in the study limitations section.

4. The methods sections suggests that fecundity data were collected during the experiment (mosquitoes allowed to lay eggs for 48 hours) but results of this assay and any potential fecundity costs of parasitism are not presented in the paper.

Thank you for this observation. Indeed, fecundity data were collected during the experiment. However, we chose not to include these results in the main manuscript as the fecundity cost of *Vavraia culicis* infection in *Anopheles gambiae*—using the same infection protocol and dose—has already been extensively reported in the literature, including studies we cite in the current manuscript. Our aim was to keep the focus on resource dynamics, which already present a complex and data-rich narrative for the reader. That said, we understand the value of transparency and completeness. We would be happy to include a brief mention of the fecundity results in the Results section and/or provide the data in the supplementary material, showing that our findings are consistent with previous reports. We hope this approach appropriately balances clarity and completeness while maintaining focus on the central contributions of the study.

Follow up: I would either briefly mention the results of the fecundity assay in the paper and provide the data as a supplementary file or remove mention of this assay from the paper altogether.

6. The resource ratio data in fig. 3 appears to combine data across blood feeding and parasitism statuses. As these are key predictor variables in the study, the expectation would be that the resource ratio data be assessed for the influence of both variables. The rationale for the choice to combine the data should be addressed in the paper.

As with Fig. 2, we initially tested the effects of blood feeding, infection status, and alive/dead status on the resource ratio. However, blood feeding and infection status showed no significant or only marginal effects, so we chose to present only the most biologically relevant and statistically significant factor—alive vs. dead—to streamline the results. Full statistical models, including all tested interactions, are provided in the Supplementary Information. Where any additional significant effects were found, we included corresponding supplementary figures. This was a deliberate balance between comprehensiveness and clarity, avoiding complex, non-informative four-way interactions that may confuse readers without adding biological insight.

Follow up: this should be clarified in the relevant section of the text when introducing this analysis

9. As the results section follows the introduction rather than the methods section, some explanation of the study design and treatments would be useful. Staring with “Mosquitoes that were not killed lived” (line 66) is a little confusing without description of the treatments. There are also inconsistencies with the naming conventions for some treatments throughout the manuscript with terms like “live”, “dead” and “killed” used, which could be confusing to some readers. I suggest picking two clear treatment names and only using these terms.

As suggested by both reviewers, we have now included a visual summary of the experimental design (now Fig. 1) before the Results section. Additionally, we introduced short explanatory paragraphs at the beginning of each Results subsection outlining the relevant treatments, sampling scheme, and study

rationale. To improve clarity and consistency, we also revised terminology throughout the manuscript to use only two terms—"alive" and "dead"—for outcome groups, replacing previously inconsistent terms such as "live" or "killed." We hope these changes make the study structure and results easier to follow.

Follow up: Reading through the manuscript, there are still references to "killed" and "live" treatments, particularly in the results section. I recommend further review of the text to resolve this issue.

Additional comments based on the revised text:

18. Fig. 4 legend – the dead dataset is noted as being represented by the color purple rather than green as it is in the figure.

19. Fig. 4 legend – Please check sample sizes are assigned to the correct treatments as there appear to be different numbers of dots than specified by the numbers currently noted in the figure (i.e., in panel a, 7 dpe, there are more than 39 pink dots and fewer than 90 green dots).

20. Lines 203-206: "Our findings show complex relationships between some ecological factors that the main malaria vector, *An. gambiae*, is commonly exposed to (blood meal and parasitism) and longevity, which is a fundamental life-history trait determining the mosquito's fitness and vectorial capacity. These results also suggest that shifts in resource allocation may underlie the observed patterns." – It would be useful to specifically define these patterns in the text.

21. Lines 227-228 - "Given that mosquitoes had continuous access to a 6% sucrose solution, we expected them to primarily use carbohydrates for energy until their reserves were depleted." – If they have unlimited access to a carbohydrate source, how would their carbohydrate reserves become depleted? Is the greater pressure not on the protein where there were two input (carryover from juvenile stages and the single blood meal) – I suggest rephrasing this part of the text.

22. Lines 273-274 – "Lipids are harder to access by the *Plasmodium* during the early stages, as they are mostly stored in the ovaries. However, once the reproductive cycle is complete, lipids become much more accessible." – These two points should be supported by references and the text should be clarified. Aren't there lipid supplies in the fat body as well as in lipid droplets associated with cell membranes and lipid transporting compounds that might also be available to *plasmodium*?

THE UNIVERSITY OF BRITISH COLUMBIA

Zoology

Faculty of Science

Luis M. Silva
Postdoctoral fellow
Department of Zoology
University of British
Columbia
Life Sciences Institute
2350 Health Sciences Mall
Vancouver, BC Canada
V6T 1Z3
E-mail: luis.silva@ubc.ca

July 2nd, 2025

Subject: Reply to reviewers' commentaries on manuscript with tracking ID COMMSBIO-24-8818

Dear Editor and reviewers,

We thank you for the comments and feedback on the manuscript. We address all the questions and comments below (reviewers' text is underlined, and our response is in italics). To facilitate the review process, we have also attached a version of the manuscript with the newly written or changed sections highlighted in yellow. We hope you will find the current documents suitable for publication.

Reviewer #1

In this study, Zeferino et al. examined how blood meal and infection by *Vavraia culicis* affect mosquito (*An. gambiae*) longevity and resource kinetics. They found that the interaction of both stressors has a cost on lifespan. Interestingly, surviving mosquitoes efficiently shift from using carbohydrates to lipids for energy, a trait missing in those that die naturally. This might suggest that energy metabolism plays a key role in survival, highlighting the intricate evolutionary relationship between hosts and parasites.

Strengths: The authors encompassed in their analysis a large range of adult mosquito lifespan, and analysed also dead individuals, which added novelty in the study. Moreover, the experimental design is well controlled.

Limitation: As the authors disclose themselves, it would be interesting to explore if these findings hold also when the blood meal is infected with malaria parasites.

We would like to sincerely thank the reviewer for their thoughtful and encouraging summary of our work. We truly appreciate your recognition of the study's strengths, including the broad analysis of mosquito lifespan and the inclusion of individuals that died naturally, as well as your comments on the relevance

of energy metabolism in the context of host–parasite interactions. Your input is both insightful and motivating.

Please find below our detailed, point-by-point responses to your comments.

General comments

1) Title and abstract

The authors discuss the hypothesis that the energetic shift observed in dead versus collected alive mosquitoes might be a predictive parameter for mortality/longevity (lines 60-61, and 192-202). However, the title states “Energetic shifts predict the mortality of Anopheles gambiae”, and in the abstract we read “shifts in carbohydrate-to-lipid ratios predict survival likelihood”. In my understanding, the word “predict” implicates to obtain specific predictive values through modeling work, with the probable need to use data from independent experiments, breakdown in training and testing datasets. Moreover, experimentally it would interesting to analyze not dead but close-to-death mosquitoes (cited by the authors at lines 195-196) for predictive purposes, as one cannot exclude the drop in changes levels happens immediately after death. The data analyzed in this manuscript show association between survival status and energetic shifts, however in my understanding no modeling work was done to clarify prediction values of the different resource readouts. Although it is interesting to discuss this hypothesis, in the absence of further matematical analysis, the word “predict” in the title (line 1) and in the abstract (line 18) should be replace “are associated with” or similar, to align the claim with the data shown. Lines 20-22. The authors put their results in the context of the current knowledge about Plasmodium dynamics in the Anopheles host. However, no Plasmodium experiments were carried out in the present manuscript. I think these are interesting hypothesis which deserve to be examined in the discussion section, but misleading to be reported on the abstract. I suggest to remove this sentence and to stick to introduction-rational-method-results-conclusion structure in the abstract.

Thank you very much for this thoughtful and constructive comment. We fully agree with your assessment that our current data show an association, rather than a formal prediction of survival outcomes, as no modeling or validation with independent datasets was conducted, a point that led to several discussions between co-authors during manuscript preparation, and we appreciate your clear articulation of the issue. Following your recommendation, we have revised both the title and abstract to replace “predict” with language more accurately reflecting the nature of our findings. We now use phrasing such as “are associated with” or “reflect” which we feel better aligns with our data. Additionally, the speculative mention of Plasmodium dynamics has been removed from the abstract and is now limited to the discussion section, where it is clearly framed as a hypothesis for future work. We appreciate your guidance in keeping the abstract focused on the core findings of the study.

2) Results section

I have a couple of suggestion to improve readability of the results section. The section starts directly into the results details. I think it would help the reader to quickly recap on the main aim and experimlent design before

that. In particular, I don't mean the details on how the experiment was done (which is extensively found in the methods) but summarize the choices of infection time (during larval development), blood feeding time, egg laying, and time points of collections for both alive and dead mosquitoes, related to the questions asked. This would provide a more straightforward understanding of the figures and result sections. Moreover, for the same reasons, I also suggest to add a conclusion paragraph to each result subsection, to help delineating the major findings, before they are discussed in details in the following section.

We thank the reviewer for this very helpful suggestion. To improve the readability and flow of the Results section, we have added a visual summary of the experimental design (the new Fig. 1) immediately before the Results. In addition, we now include a brief introductory paragraph at the beginning of each Results subsection that outlines the experimental logic, including sampling scheme and how it relates to the study questions. We have also added a short conclusion paragraph to the end of each Results subsection to clearly highlight the major findings prior to their integration in the Discussion. We hope these changes make the structure and interpretation of our results more accessible to readers.

3) Figures

The authors can consider indicating the sample sizes directly in the graphs, below the corresponding dotplot column, for Fig. 1A and Fig. 2. It would lighten and increase readability of the figure legends. For Fig. 2 and 3, alive and dead individuals are compared side by side; however, while live samples correspond to the indicated time point in the axis, dead individuals were pooled for +/- 2.5 around the indicated time point (as explained in details in the methods, line 314). I think it's important to mention this point also in the corresponding figure legend. In Fig. 3, I suggest to choose two colors that differ more between each other, for the "dead" vs "alive" comparisons; I personally find the graph difficult to read with the current colors.

We thank the reviewer for these spot-on suggestions to improve the clarity and readability of our figures. We have now indicated the sample sizes directly below the corresponding dot plots in Figures 1A (now 2a) and 2 (now 3), and added a note to the legends of Figures 2 (now 3) and 3 (now 4) clarifying that data from dead individuals were pooled over a ± 2.5 day window around each time point. Additionally, we updated the colour scheme in Figure 4 to enhance visual contrast between dead and alive samples, making the graph easier to interpret. Together with the adjustments made in response to the previous comment, we believe these changes greatly improve the accessibility and flow of the results section.

Specific comments:

4) Line 52. Citation 19 refers to another mosquito, *A. aegypti*; please make it clear in the text to avoid confusions.

*Thank you for pointing this out. We have clarified in the revised text that the findings in citation 19 refer specifically to *Aedes aegypti* to avoid confusion.*

5) Lines 58-59. The authors state "Our findings show that the resource levels throughout the adult mosquito's life are influenced by blood meal and infection". Moreover, the authors state at lines 154-156 "In particular, the resource levels throughout the adult mosquito's life were influenced by blood meal and by the infection by a

microsporidian. In addition, at lines 207-209, it reads “By adulthood, infected mosquitoes exhibit resource depletion of 2-3 times higher than their uninfected counterparts (e.g., carbohydrates: 64 vs. 187 ng/μL; lipids: 20 vs. 48 ng/μL by day 7). However, in my understanding, resource content was similar in blood-fed and unfed mosquitoes and was independent of the infection status, or combination of the treatments (Table S2), as also mentioned in the results (lines 95-98, 102-105, and 110-113). I might have misunderstood some point, but this seems in apparent contradiction to me. Please clarify / harmonize throughout the text.

Thank you for this valuable observation. We agree that the original statements were inconsistent with the results presented in the Results section. We have revised the text throughout the manuscript to harmonize the interpretation and avoid suggesting a direct influence of blood meal or infection on overall resource levels.

Instead, we now emphasize that although resource levels were not significantly affected by blood meal or infection status, they appear to reflect the likelihood of survival. Specifically, in the Discussion section, we now highlight that differences in lipid content (as long-term reserves) between infected and uninfected individuals suggest that infection may lead to a depletion of stored energy. In contrast, carbohydrate levels (short-term reserves) did not differ as markedly, which may indicate that infected individuals were able to compensate for potential depletion of this resource type by feeding on the carbohydrate source provided. We hope these clarifications resolve the apparent contradiction and improve the coherence of our interpretation.

6) Line 60. The authors state “resources were tightly linked to adult survivorship”. I am not sure I understand this sentence: (some) resource levels? (specific) resource shifts? Please specify.

We have revised the sentence to clarify that it refers specifically to carbohydrates and lipids. These two resource types appear to be tightly linked to adult survivorship, with lower levels generally associated with increased mortality risk. This specification has been added to improve clarity.

7) Line 61. The word “understanding” is repeated twice in the same sentence, please rephrase for clarity.

We have rephrased the sentence for improved clarity and readability.

8) Lines 76-77. Since collected alive individuals were excluded from this graph (Fig. A), please remove the “alive” “dead” legend in the right lower corner of the figure, as it makes the reader expecting to see also live mosquitoes plotted in the graphs.

The “alive”/“dead” legend has been removed from the graph to avoid confusion and better reflect the data presented.

9) Lines 85-87. The authors state “Spore load ... was higher in dead than alive mosquitoes, independently of their age”. This concept is also taken back in the discussion at lines 160-162. Although the statistical significance of this result is reported in the results and shown in Table S2, it could help to show an additional graph with

spore load in dead vs alive mosquitoes all timepoints merged, for the reader to visually appreciate cumulative effect size.

A new supplementary figure (Fig. S1) has been added showing spore load in dead vs. alive mosquitoes with data pooled across all timepoints, allowing for easier visual appreciation of the cumulative effect size.

10) Lines 116-117. Please clarify between which groups the multiple comparisons were applied. It is not clear to me because the strait line in the graphs spans over two groups.

We have clarified in the figure legend that the multiple comparisons were performed between alive and dead individuals. For this analysis, the "blood meal" and "no blood meal" treatments were grouped together, as blood meal had no significant effect on any of the measured variables.

11) Line 153-154. The authors state "They suggest the role of resource allocation underlying these relationship." I am not sure I understand. Do the mean "They suggest that resource allocation plays a role underlying these relationships"? Please clarify.

We agree that the sentence was unclear and have now rephrased it. The revised sentence more clearly conveys our intended meaning, which is that our findings imply a role for resource allocation in shaping the relationships between ecological factors and longevity.

12) Line 157. The author states "Taking a blood meal strongly shortened the mosquitoes' lives". However, accordingly to Figure 1a the reduction corresponds only to about 3.3 days on a mean lifespan of 18, and is true only when comparing within uninfected mosquitoes. "Strongly shortened" seems overstating to me, please rephrase.

We initially used "strongly" because a reduction of approximately 3.3 days on a mean lifespan of ~20 days represents about a 15% decrease in longevity, which we considered biologically meaningful. However, we agree that the term may overstate the effect, especially given the context of the sentence and the fact that it applies only to uninfected mosquitoes. As it does not add critical information to the argument being made, we have removed "strongly" from the sentence to improve clarity and precision.

13) Lines 158-159. The authors state "In our design, individuals had ad libitum access to blood", however in the methods is reported that single mosquitoes were left to feed one time for 5 minutes (line 254). I think "ad libitum" is misleading, as it's normally intended for as much or as often as desired. Please rephrase, for example by stating mosquitoes were fully engorged, or similar.

You are correct that "ad libitum" was misleading in this context. Our intention was to indicate that mosquitoes were allowed to feed without restriction during the 5-minute period, which is typically sufficient for them to fully engorge. To clarify this, we have revised the sentence.

14) Lines 179-183. The authors state “Indeed, this energetic shift interestingly coincides with an expected production of a new cluster of eggs.”. I haven’t found though any data showing a new (second) production of eggs in the results. Please rephrase to highlight this is an hypothesis. Moreover, the authors state “It has been discussed that several mosquito species do not need the energetic resources to produce the new cluster of eggs, but instead solely the iron signalling. However, in the citations mentioned I only found this related to ticks and kissing bugs. Please cite specifically in which mosquitoes this has been discussed or replace with (anautozenous) arthropods. Moreover, assuming that is the case for A. gambiae studied here, how do the authors imagine the mosquitoes activate (again) iron signaling cascade in this experimental setting in the absence of a (second) blood meal? Please revise carefully this paragraph or remove it.

We agree that the original phrasing of the sentence could misleadingly suggest we directly measured a second egg production event. We have now rephrased this part of the text to clarify that it is a hypothesis based on the timing of physiological changes, rather than a result from our data. Regarding the second point, while the cited literature indeed includes ticks and kissing bugs, it also discusses this mechanism in mosquitoes such as Aedes aegypti and Anopheles gambiae. We have revised the sentence to more accurately reflect this. Finally, we have expanded this paragraph to more clearly explain the logic of our hypothesis and the potential role of iron signalling, while acknowledging the limitations of our experimental setup in fully testing this mechanism.

15) Line 204. Citation 29 refers to A. aegypti larvae, please state this also in the text to clarify there is this additional differences compared to the data shown in this manuscript.

Thank you for the suggestion. We have revised the sentence to clearly specify that the referenced study was conducted in Aedes aegypti larvae, and we now explicitly contrast it with our findings in Anopheles gambiae adults.

16) Line 216. The authors state “This result was shown to be due to the need for lipids by sporozoites but not oocysts.”, referring to citation 30. In this reference, the authors have shown that neutral lipids are need to form virulent sporozoites and acquired during sporulation (formation of sporozoites) happening in the oocyst. In the P. berghei model, a mosquito lipid transporter starts to accumulate around growing oocysts from D7 onward (Fig S3). Indeed by restoring resources through an additional blood meal at D7 post infection, the sporulation can be rescued (Fig. 4). In this context the sentence cited above results imprecise, please rephrase.

Very good observation, thank you. The sentence has been corrected and we hope is now adequate.

17) Line 244. The authors state that each experiment “Each experiment was started with eggs from three cages”. How many experiments were performed? Were all conditions repeated in all experiments? Sample sizes per each group are indicated clearly in the figure legends, but I didn’t find any information about experimental replicates.

All data presented in the main text were collected from a single, large-scale experiment. The phrasing “each experiment was started with eggs from three cages” was misleading in this context and reflects

a standard lab protocol rather than multiple experimental replicates. We have corrected the sentence in the manuscript to avoid confusion. Additionally, we clarify that a separate pilot experiment, presented only in the supplementary materials, was performed to assess resource degradation over time. It was not used for statistical analysis of the main results.

18) Line 250. How many individual mosquitoes were hosted per cage?

Each cage initially housed approximately 200 individual mosquitoes. As about half of these were males, which we later removed and replaced with untreated males, we have rephrased this section in the manuscript to improve clarity.

19) Line 254. Please replace "on TGZ's arm" with "on human skin (TGZ)", for clarity. Were mosquitoes that didn't take a blood meal excluded or used in the unfed group? Where unfed mosquitoes also moved to the egg laying cup, and later on to a normal cup again, for a mock control? Please specify.

We have replaced "on TGZ's arm" with "on human skin (TGZ)" for clarity, as recommended. Mosquitoes that did not take a blood meal were excluded from the experiment. Additionally, unfed mosquitoes were not moved to the egg-laying cups, as we did not perform a mock handling procedure for this group. We have revised the manuscript accordingly to clarify these points.

Reviewer #2

In their study 'Energetic shifts predict the mortality of Anopheles gambiae' the authors examine the effects of two stressors (blood feeding and infection with a microsporidian parasite) impact energy dynamics in the malaria mosquito Anopheles gambiae. They seek to examine how specific resources and resource ratios in this mosquito are impacted by these stressors across a range of age points, and for mosquitoes that die naturally or are killed and examined at those age points. Mosquitoes are either parasitized as larvae or not, and/or offered a chance to blood feed at 7 days post eclosion, or not. Levels of parasite spores (if infected) were quantified using a hemocytometer, while levels of proteins, carbohydrates, and lipids were quantified via colorimetric assay. The authors observe key differences in carbohydrate and lipids levels and parasite load with age, and also between mosquitoes that died naturally or were live sampled during the experiment. The findings of the study are novel and interesting and should open the way to broader investigations of interconnectivity between metabolism, immunity, and physiology in insects. However, the framing of the research is much broader than the actual experiments that were performed and there are multiple subjects in the description of the experimental design that need further clarification.

We would like to sincerely thank the reviewer for their thoughtful and constructive feedback on our manuscript. We are especially grateful for your recognition of the novelty and potential broader implications of our findings, particularly regarding the links between metabolism, immunity, and physiology in insects. Your comments are highly encouraging and align with our broader aim of contributing to this growing field. We also appreciate your observations regarding the framing of the manuscript and the need for greater clarity in certain aspects of the experimental design. We have

carefully revised the text to better align our framing with the scope of the experiments conducted and to clarify methodological details where needed. Please find below our detailed, point-by-point responses to each of your comments.

Major comments

1. The abstract and parts of the discussion mention adaptation of Plasmodium to mosquito host resource availability, but no Plasmodium model is used in this study and findings based on a single microsporidian strain are not necessarily transferrable to other, potentially more pathogen microsporidians, let alone a distinct clade of parasites like Plasmodium.

We fully agree that extrapolating our findings to Plasmodium dynamics was speculative, especially in the absence of a Plasmodium model in the present study. This concern was also raised by Reviewer 1, and we appreciate the opportunity to clarify our position. As a result, we have removed the mention of Plasmodium from the abstract to keep the focus on the experimental system actually used and avoid overstating the implications of our findings. However, we believe the potential relevance of resource dynamics in shaping interactions with other parasites—including Plasmodium—remains an interesting avenue for future research. Therefore, we have retained a brief mention in the discussion, where it is clearly framed as a hypothesis and not as a direct implication of our results. We hope this revision appropriately addresses your concern and improves the clarity and focus of the manuscript.

2. It's difficult to gauge the effects of blood feeding in this experiment. Mosquitoes were offered a single blood meal at 7 days post-eclosion. It is unclear whether they were then screened to determine feeding status (not fed, partially fed, or fully fed). Resource levels were then quantified at 7 dpe (pre-blood feeding), or at 12 dpe, which would be after most mosquitoes had finished digestion of their blood meal and blood meal-derived resources were catabolized and stored. In nature, over the course of a lifespan of approximately 30 days, it would be expected that a female mosquito would blood feed several times.

In our design, mosquitoes were offered a single blood meal at 7 days post-eclosion, and only fully engorged individuals were retained in the experiment. This has now been clarified in the methods section to avoid confusion regarding feeding status. Regarding resource quantification, we chose 12 days post-eclosion as our primary time point to allow sufficient time for digestion and for blood meal-derived resources to be metabolized and either stored or utilized. While this means circulating nutrients from the blood meal were no longer directly measurable, we were able to assess stored metabolic resources—particularly lipids and carbohydrates—which reflect the physiological impact of blood feeding. Proteins, which may be more quickly allocated to reproductive or metabolic processes, are naturally more variable at this later time point. We agree that a more frequent sampling regime—such as daily measurements—would have provided a more accurate insight into the dynamics of resource metabolism post-blood meal. However, due to logistical constraints, such as the need to maintain a full factorial design across three experimental factors (infection status, blood feeding status, and survival status), it was not feasible to include additional time points or offer repeated blood meals. The factorial design inherently required a large number of mosquitoes, and ensuring sufficient sample sizes—

especially for individuals that died naturally—necessitated prioritizing critical time points over high resolution. Adding extra blood meals or spreading data collection across different experimental batches would have further complicated interpretation, particularly given the sensitivity of metabolic measurements to environmental and temporal variation.

We hope this explanation clarifies the rationale and experimental constraints that shaped our design decisions and the organization of the study.

3. From the description in the methods, male and female mosquitoes were allowed time to mate prior to blood feeding. From the manuscript, it is unclear whether all female mosquitoes had mated successfully and produced eggs. It is reasonable to hypothesize that energy dynamics would differ between gravid and non-gravid females, and that this might also have been impacted by parasitism with *V. culicis*.

*We agree that mating status and subsequent reproductive activity can influence energy dynamics and that parasitism might modulate these relationships. In our design, females were left with males for seven days prior to blood feeding. Based on data we have collected (not shown in this manuscript), we know that this duration results in nearly 100% of females being motivated to blood feed—a behavior tightly linked to successful insemination in *Anopheles* mosquitoes. To further ensure this, we excluded from the experiment any females that did not take a full blood meal, which likely eliminated most unmated individuals. While we acknowledge that this approach could have removed individuals potentially affected by the parasite (e.g., if infection reduced mating success), it was necessary to maintain consistency across treatment groups. Additionally, we recorded whether females laid eggs after blood feeding, and we explored potential differences in resource levels based on this variable. However, since egg-laying status did not show a detectable effect on energetic profiles, and in light of model complexity limitations (too many factors and interactions preventing model convergence), we opted not to include it in the main analyses. We agree with the reviewer that a more detailed investigation into the differences in energy dynamics between gravid and non-gravid females, especially in interaction with parasitism, would be a valuable direction for future work. However, once again, logistics associated with sample size and the amount of combinatorial treatments prevent us from exploring that direction.*

4. The methods sections suggests that fecundity data were collected during the experiment (mosquitoes allowed to lay eggs for 48 hours) but results of this assay and any potential fecundity costs of parasitism are not presented in the paper.

*Thank you for this observation. Indeed, fecundity data were collected during the experiment. However, we chose not to include these results in the main manuscript as the fecundity cost of *Vavraia culicis* infection in *Anopheles gambiae*—using the same infection protocol and dose—has already been extensively reported in the literature, including studies we cite in the current manuscript. Our aim was to keep the focus on resource dynamics, which already present a complex and data-rich narrative for the reader. That said, we understand the value of transparency and completeness. We would be happy*

to include a brief mention of the fecundity results in the Results section and/or provide the data in the supplementary material, showing that our findings are consistent with previous reports. We hope this approach appropriately balances clarity and completeness while maintaining focus on the central contributions of the study.

5. Other details in the methods and study design that need clarification or adjustment:

- Age of mosquitoes should be described in relation to time post-hatch for juveniles and time post-eclosion for adults, not in terms of “days old”.

We agree that specifying post-hatch and post-emergence can improve biological precision. However, we chose to use “days old larvae” or “days old adults” consistently throughout the manuscript to ensure accessibility for a broad readership, including those less familiar with mosquito development terminology. Given that the majority of our data focuses on adult mosquitoes and the age-related dynamics of resource usage and survival, we believe this terminology sufficiently conveys the relevant biological timeline. We have ensured that the context is always clear (i.e., whether referring to larvae or adults), and for readability and clarity, we would prefer to retain the current phrasing.

- There is no confirmation whether the amount of blood imbibed by parasitized mosquitoes differed from non-parasitized mosquitoes.

*In this experiment, we did not measure blood meal volume due to logistical constraints, including the large sample size required by the factorial design. However, based on previous unpublished data from our lab, we know that *Vavraia culicis* infection does not alter the amount of blood ingested by *Anopheles gambiae* females. It may, however, increase blood-feeding motivation, particularly in unmated females. Additionally, due to *Anopheles*' ability to perform post-feeding diuresis, any subtle differences in blood volume are likely minimized, especially given the large 5-minute feeding window used here. All mosquitoes in our study were fully engorged, likely because of the time they were allowed to blood feed, making it unlikely that feeding volume differences are present.*

- It is unclear whether the experimental design was run just once or if it was repeated.

Thank you for highlighting this ambiguity, also noted by Reviewer 1. We clarify that the main experiment was conducted as a single experimental run and this has now been clearly stated in the Methods section.

- Details on how the longevity assay was performed are not provided in the methods.

We have now included a dedicated and expanded description of the longevity assay protocol in the revised Methods section.

- It is unclear whether the measurement of resource quantities for each mosquito specimen were made once or taken as the average of duplicate or triplicate measurements.

While the information regarding replicates was already included in the original Methods section, we recognize that it may not have been sufficiently clear. We have now revised and rephrased the relevant paragraph to make it more explicit.

6. The resource ratio data in fig. 3 appears to combine data across blood feeding and parasitism statuses. As these are key predictor variables in the study, the expectation would be that the resource ratio data be assessed for the influence of both variables. The rationale for the choice to combine the data should be addressed in the paper.

As with Fig. 2, we initially tested the effects of blood feeding, infection status, and alive/dead status on the resource ratio. However, blood feeding and infection status showed no significant or only marginal effects, so we chose to present only the most biologically relevant and statistically significant factor—alive vs. dead—to streamline the results. Full statistical models, including all tested interactions, are provided in the Supplementary Information. Where any additional significant effects were found, we included corresponding supplementary figures. This was a deliberate balance between comprehensiveness and clarity, avoiding complex, non-informative four-way interactions that may confuse readers without adding biological insight.

7. Longevity data are typically non-linear, as death rates in insect populations are inconstant over time. As such, it is unusual to see linear models used in their analysis. For Figure 1a, where data were collected every 12 hours, this should be an XY plot displaying the percentage survival over time rather than a dot plot. Additionally, the Y-axis depicts age at death rather than mean age at death, which warrants a change in the axis title.

We agree that survival analysis is often preferred when mortality dynamics differ strongly between groups, especially when hazards are non-proportional. We considered both approaches—survival analysis and age-at-death—and found that mortality was consistent and normally distributed within groups, with no censoring. Therefore, we opted for a more intuitive representation: age at death, analyzed with linear models, which simplifies interpretation while remaining statistically sound.

The Y-axis in Fig. 1a does represent individual age at death, while we also include the mean and standard errors of the mean in each group, and we have now clarified this in the legend. We chose to use individual dots to allow readers to visually assess variance and identify any unusual mortality patterns (e.g., bimodality), which were not observed. While a survival curve could have been used, we believe the chosen format better highlights group differences in a straightforward and reader-friendly way.

Minor comments

8. The abstract and introduction highlight several limitations of study design for projects assessing energy dynamics in insects, and the final paragraph of the introduction highlights the authors “comprehensive approach” to addressing interactions between stressors, longevity, and resource levels. Given the very broad scale and

scope of these interactions and what is possible to achieve in one experimental design, a more neutral choice of language would be appropriate here.

Thank you for your suggestion. We have now changed the abstract, introduction, and parts of the discussion, and we hope it is clearer to the reader.

9. As the results section follows the introduction rather than the methods section, some explanation of the study design and treatments would be useful. Staring with “Mosquitoes that were not killed lived” (line 66) is a little confusing without description of the treatments. There are also inconsistencies with the naming conventions for some treatments throughout the manuscript with terms like “live”, “dead” and “killed” used, which could be confusing to some readers. I suggest picking two clear treatment names and only using these terms.

As suggested by both reviewers, we have now included a visual summary of the experimental design (now Fig. 1) before the Results section. Additionally, we introduced short explanatory paragraphs at the beginning of each Results subsection outlining the relevant treatments, sampling scheme, and study rationale. To improve clarity and consistency, we also revised terminology throughout the manuscript to use only two terms—“alive” and “dead”—for outcome groups, replacing previously inconsistent terms such as “live” or “killed.” We hope these changes make the study structure and results easier to follow.

10. There are lots of 100 values in the carbohydrate and lipid data. Is it possible that the lower limit of the standard curve was too high or the limit of detection/dilution factor too low?

The protocols were carefully optimized over several weeks prior to the experiment to ensure a balance between sensitivity and accuracy. Each plate included an 8-point standard curve to cover a wide range of optical density (OD) values. The vast majority of samples fell within the 0–1 OD range, with only two exceptions slightly exceeding these bounds. Therefore, we are confident that the standard curves were appropriate and robust, and that the values are reliable. We believe the “100” values do not represent ceiling effects from curve limitations but rather reflect genuine biological variation.

11. Figs 1a and 1b – the key depicts filled circles for live mosquitoes, but it appears that no filled circles appear in these two plots.

As it was also pointed out by the first reviewer, the “alive”/“dead” legend has been removed from the graph to avoid confusion and better reflect the data presented.

12. Large portions of the Fig. 2 and 3 legends are a list of sample sizes. Is there a more effective way to provide this information, such as by providing a range of sample sizes per treatment on the figure and then providing the raw data as a supplementary file?

As also suggested by Reviewer 1, we have updated Figures 2 and 3 to display sample sizes directly within the figures.

13. There is a disparity with information on blood feeding in the manuscript. In the methods it states that there was one five minute blood feeding opportunity, while in the discussion it states that mosquitoes had ad libitum access to blood. This should be clarified.

Thank you for catching this inconsistency. The phrase “ad libitum” was misleading and has now been removed. Mosquitoes were given unrestricted access to blood for a fixed period of 5 minutes, which is sufficient for full engorgement in Anopheles. This has been clarified in the revised manuscript.

14. There is a disparity in how the longevity cost of V. culicis is described. In the introduction, it states that there is minimal effect on longevity, while in the discussion, the authors note that their findings and the findings of other studies highlight significant shortening of lifespan. This should be clarified.

We agree with the reviewer that the statements in the Introduction and Discussion may appear contradictory. In the Introduction, we emphasize that V. culicis has limited effects on host longevity, based on previous studies and our rationale for using it to track long-term changes in resource allocation. However, our results do show a statistically significant reduction in lifespan—specifically, a 3.3-day decrease on a lifespan of approximately 20 days. As noted by Reviewer 1, this is a relatively small reduction and aligns with previous findings (e.g., Silva and Koella 2024; Agnew and Koella 1997). We have now revised both the Introduction and Discussion to clarify this point: while V. culicis reduces longevity to some extent, the effect is moderate and does not substantially limit the ability to follow physiological dynamics over time. We hope this clarifies the consistency between our findings and existing literature.

15. Lines 165-166 – “Indeed, at many ages, the resource content was lower in the mosquitoes that died naturally than in those we killed” - for some ages resource content was lower but at others it was higher. The results are more complicated than made out in this statement.

We agree with the reviewer that this statement oversimplifies the results and does not fully reflect the observed complexity. As it also did not add substantial value to the paragraph, we have removed it from the revised manuscript.

16. Line 178 – “likely due to the inability to reach their sucrose source” – this needs to be explained a bit better – was the sucrose moved or were the mosquitoes less mobile with age? Is there evidence to support a change in mobility in your experiment?

We have now expanded the explanation to clarify that the “inability to reach the sucrose source” refers to decreased mobility with age, which we observed qualitatively during the experiment. This hypothesis aligns with the timing of the observed shift from carbohydrate to lipid use, although we acknowledge that we did not directly measure locomotion.

17. Line 183 – “We show support for this hypothesis with a consistent decrease in” – it’s not a consistent decrease if resource levels then increase again for older mosquitoes. This should be rephrased.

We agree that the term “consistent decrease” could be misleading given the recovery observed in later time points. We have revised this part of the manuscript to clarify that the carbohydrate content shows a progressive decrease until day 22, after which it partially recovers—a pattern that supports our hypothesis regarding the timing of energy allocation for egg development and replenishment from sucrose intake.

Yours sincerely,

Luís M. Silva

(on behalf of all authors)

THE UNIVERSITY OF BRITISH COLUMBIA

Zoology

Faculty of Science

Luis M. Silva
Postdoctoral fellow
Department of Zoology
University of British
Columbia
Life Sciences Institute
2350 Health Sciences Mall
Vancouver, BC Canada
V6T 1Z3
E-mail: luis.silva@ubc.ca

October 1st, 2025

Subject: Reply to reviewers' commentaries on manuscript with tracking ID COMMSBIO-24-8818

Dear Editor and reviewers,

We thank you for the positive and constructive feedback. We have addressed the remaining comments below in *italic*. We hope that the manuscript is now adequate for publication.

Reviewer #2

Response to previous comments:

Previous comment by reviewer: 2. It's difficult to gauge the effects of blood feeding in this experiment. Mosquitoes were offered a single blood meal at 7 days post-eclosion. It is unclear whether they were then screened to determine feeding status (not fed, partially fed, or fully fed). Resource levels were then quantified at 7 dpe (pre-blood feeding), or at 12 dpe, which would be after most mosquitoes had finished digestion of their blood meal and blood meal-derived resources were catabolized and stored. In nature, over the course of a lifespan of approximately 30 days, it would be expected that a female mosquito would blood feed several times.

Our previous reply: In our design, mosquitoes were offered a single blood meal at 7 days post-eclosion, and only fully engorged individuals were retained in the experiment. This has now been clarified in the methods section to avoid confusion regarding feeding status. Regarding resource quantification, we chose 12 days post-eclosion as our primary time point to allow sufficient time for digestion and for blood meal-derived resources to be metabolized and either stored or utilized. While this means circulating nutrients from the blood meal were no longer directly measurable, we were able to assess stored metabolic resources—particularly lipids and carbohydrates—which reflect the physiological impact of blood feeding. Proteins, which may be more quickly allocated to reproductive or metabolic processes, are naturally more variable at this later time point. We agree that a more frequent sampling regime—such as daily measurements—would have provided a more accurate insight into the dynamics of resource metabolism post-blood meal. However, due to logistical constraints, such as the need to maintain a full factorial design across three experimental factors (infection

status, blood feeding status, and survival status), it was not feasible to include additional time points or offer repeated blood meals. The factorial design inherently required a large number of mosquitoes, and ensuring sufficient sample sizes—especially for individuals that died naturally—necessitated prioritizing critical time points over high resolution. Adding extra blood meals or spreading data collection across different experimental batches would have further complicated interpretation, particularly given the sensitivity of metabolic measurements to environmental and temporal variation. We hope this explanation clarifies the rationale and experimental constraints that shaped our design decisions and the organization of the study.

Follow-up comment by the reviewer: I agree with what you have said in your response that it would have been impractical to repeat the metabolite quantification for additional time points. To clarify the choice of a 12 dpe time point, it would be good to add the rationale you provided in your rebuttal to the methods section. Additionally, my original point should be addressed in the study limitations section.

Current reply: Thank you for your feedback. As suggested, we included the reasoning for having only one blood-feeding time-point in the Materials and Methods (lines 370-375 & 382-385). We also included it as one of the limitations of the study in the discussion of the findings (lines 270-276).

Previous comment from reviewer: 4. The methods sections suggests that fecundity data were collected during the experiment (mosquitoes allowed to lay eggs for 48 hours) but results of this assay and any potential fecundity costs of parasitism are not presented in the paper.

Our previous reply: Thank you for this observation. Indeed, fecundity data were collected during the experiment. However, we chose not to include these results in the main manuscript as the fecundity cost of *Vavraia culicis* infection in *Anopheles gambiae*—using the same infection protocol and dose—has already been extensively reported in the literature, including studies we cite in the current manuscript. Our aim was to keep the focus on resource dynamics, which already present a complex and data-rich narrative for the reader. That said, we understand the value of transparency and completeness. We would be happy to include a brief mention of the fecundity results in the Results section and/or provide the data in the supplementary material, showing that our findings are consistent with previous reports. We hope this approach appropriately balances clarity and completeness while maintaining focus on the central contributions of the study.

Follow-up comment by the reviewer: I would either briefly mention the results of the fecundity assay in the paper and provide the data as a supplementary file or remove mention of this assay from the paper altogether.

Current reply: We have clarified in the Methods section (lines 313-316) that fecundity data were collected but are not presented in this manuscript, as the fecundity cost of *V. culicis* infection in *An. gambiae* has already been extensively reported in the literature (that we cite). We also explained that our focus here is on resource dynamics, which constitute the central contribution of the study. The revised text makes this rationale explicit to ensure transparency while keeping the manuscript focused.

Previous comment from reviewer: 6. The resource ratio data in fig. 3 appears to combine data across blood feeding and parasitism statuses. As these are key predictor variables in the study, the expectation would be that the resource ratio data be assessed for the influence of both variables. The rationale for the choice to combine the data should be addressed in the paper.

Our previous reply: As with Fig. 2, we initially tested the effects of blood feeding, infection status, and alive/dead status on the resource ratio. However, blood feeding and infection status showed no significant or only marginal effects, so we chose to present only the most biologically relevant and statistically significant factor— alive vs. dead—to streamline the results. Full statistical models, including all tested interactions, are provided in the Supplementary Information. Where any additional significant effects were found, we included corresponding supplementary figures. This was a deliberate balance between comprehensiveness and clarity, avoiding complex, non-informative four-way interactions that may confuse readers without adding biological insight.

Follow-up comment by the reviewer: This should be clarified in the relevant section of the text when introducing this analysis.

***Current reply:** The section of the Materials and Methods describing this analysis has now been completed (Lines 382-385).*

Previous comment from the reviewer: 9. As the results section follows the introduction rather than the methods section, some explanation of the study design and treatments would be useful. Starting with “Mosquitoes that were not killed lived” (line 66) is a little confusing without description of the treatments. There are also inconsistencies with the naming conventions for some treatments throughout the manuscript with terms like “live”, “dead” and “killed” used, which could be confusing to some readers. I suggest picking two clear treatment names and only using these terms.

Our previous reply: As suggested by both reviewers, we have now included a visual summary of the experimental design (now Fig. 1) before the Results section. Additionally, we introduced short explanatory paragraphs at the beginning of each Results subsection outlining the relevant treatments, sampling scheme, and study rationale. To improve clarity and consistency, we also revised terminology throughout the manuscript to use only two terms—“alive” and “dead”—for outcome groups, replacing previously inconsistent terms such as “live” or “killed.” We hope these changes make the study structure and results easier to follow.

Follow-up comment by the reviewer: Reading through the manuscript, there are still references to “killed” and “live” treatments, particularly in the results section. I recommend further review of the text to resolve this issue.

***Current reply:** We apologize for missing those typos. We have now corrected the terminology in the new version of the manuscript, and we hope the manuscript is appropriate.*

Additional comments based on the revised text:

18. Fig. 4 legend – the dead dataset is noted as being represented by the color purple rather than green as it is in the figure.

Thank you for pointing that typo out. The mistake has been corrected.

19. Fig. 4 legend – Please check sample sizes are assigned to the correct treatments as there appear to be different numbers of dots than specified by the numbers currently noted in the figure (i.e., in panel a, 7 dpe, there are more than 39 pink dots and fewer than 90 green dots).

Thank you for noticing the mistake. The figure and caption are now corrected.

20. Lines 203-206: “Our findings show complex relationships between some ecological factors that the main malaria vector, *An. gambiae*, is commonly exposed to (blood meal and parasitism) and longevity, which is a fundamental life-history trait determining the mosquito’s fitness and vectorial capacity. These results also suggest that shifts in resource allocation may underlie the observed patterns.” – It would be useful to specifically define these patterns in the text specifically.

*We agree that clearly defining the observed patterns is important. In the revised version (Lines 164-170), we kept the opening paragraph of the discussion as a high-level overview of the main findings, while the subsequent paragraphs explicitly define and explore each of the patterns in detail (e.g., the effects of blood feeding and *V. culicis* infection on longevity, and the shifts in carbohydrate and lipid dynamics). We feel this structure improves readability by providing a concise introduction before moving into a more detailed treatment of each result. To address the reviewer’s comment, we have slightly rephrased the opening paragraph to make the transition to these specific discussions clearer.*

21. Lines 227-228 - “Given that mosquitoes had continuous access to a 6% sucrose solution, we expected them to primarily use carbohydrates for energy until their reserves were depleted.” – If they have unlimited access to a carbohydrate source, how would their carbohydrate reserves become depleted? Is the greater pressure not on the protein where there were two input (carryover from juvenile stages and the single blood meal) – I suggest rephrasing this part of the text.

We agree that, given continuous access to sucrose, carbohydrate depletion may at first seem counterintuitive. To clarify, we hypothesize that aging mosquitoes may experience reduced mobility, which limits their ability to reach the sucrose source. This interpretation is supported by our direct observations of mosquitoes near day 20, when individuals showed difficulty reaching the cotton balls soaked in sucrose. We therefore suggest that impaired access (even under unlimited supply) explains the net decrease in carbohydrate reserves and the subsequent metabolic shift toward lipid utilization. With respect to proteins, while they play a key role in mosquito physiology and reproduction, they are

not the primary energetic resource in this context and thus cannot account for the observed shift from carbohydrate to lipid use. For these reasons, we have slightly rephrased the sentence for clarity but retained the overall interpretation, which we believe remains well supported by our data (Lines 189-191).

22. Lines 273-274 – “Lipids are harder to access by the *Plasmodium* during the early stages, as they are mostly stored in the ovaries. However, once the reproductive cycle is complete, lipids become much more accessible.” – These two points should be supported by references and the text should be clarified. Aren't there lipid supplies in the fat body as well as in lipid droplets associated with cell membranes and lipid transporting compounds that might also be available to *Plasmodium*?

Thank you for the interesting comment. It is true that lipid dynamics and storage are a bit more complex than what we may have written. Therefore, we described it a bit further in the text to make it more comprehensive and informative for the reader. We also cited the sentences accordingly.

Yours sincerely,

Luís M. Silva

(on behalf of all authors)